# Post-translational flavinylation is associated with diverse extracytosolic redox functionalities throughout bacterial life

**Raphaël Méheust[1,2,3†], Shuo Huang[4,5†], Rafael Rivera-Lugo[6], Jillian F Banfield[1,2], Samuel H Light[4,5]***

[1]Department of Earth and Planetary Science, University of California, Berkeley, Berkeley, United States; [2]Innovative Genomics Institute, Berkeley, United States; [3]LABGeM, Génomique Métabolique, Genoscope, Institut François Jacob, CEA, Evry, France; [4]Duchossois Family Institute, University of Chicago, Chicago, United States; [5]Department of Microbiology, University of Chicago, Chicago, United States; [6]Department of Molecular and Cell Biology, University of California, Berkeley, Berkeley, United States

**Abstract** Disparate redox activities that take place beyond the bounds of the prokaryotic cell cytosol must connect to membrane or cytosolic electron pools. Proteins post-translationally flavinylated by the enzyme ApbE mediate electron transfer in several characterized extracytosolic redox systems but the breadth of functions of this modification remains unknown. Here, we present a comprehensive bioinformatic analysis of 31,910 prokaryotic genomes that provides evidence of extracytosolic ApbEs within ~50% of bacteria and the involvement of flavinylation in numerous uncharacterized biochemical processes. By mining flavinylation-associated gene clusters, we identify five protein classes responsible for transmembrane electron transfer and two domains of unknown function (DUF2271 and DUF3570) that are flavinylated by ApbE. We observe flavinylation/iron transporter gene colocalization patterns that implicate functions in iron reduction and assimilation. We find associations with characterized and uncharacterized respiratory oxidoreductases that highlight roles of flavinylation in respiratory electron transport chains. Finally, we identify interspecies gene cluster variability consistent with flavinylation/cytochrome functional redundancies and discover a class of 'multi-flavinylated proteins' that may resemble multi-heme cytochromes in facilitating longer distance electron transfer. These findings provide mechanistic insight into an important facet of bacterial physiology and establish flavinylation as a functionally diverse mediator of extracytosolic electron transfer.

*For correspondence:
samlight@uchicago.edu

†These authors contributed equally to this work

## Introduction

Essential microbial-environmental interactions take place beyond the bounds of the cytosol. In prokaryotes, extracytosolic biochemical processes can be situated at the extracytosolic side of the inner membrane, periplasm, cell wall, or surrounding environment. Within the extracytosolic space, redox reactions represent an important class of activities with functions in respiration, the maintenance/repair of extracytosolic proteins, and the assimilation of minerals (*Bertini et al., 2006*; *Cho and Collet, 2013*; *Schröder et al., 2003*).

Extracytosolic redox processes require electron transfer pathways that connect with electron pools in the cytosol or membrane. Membrane proteins transfer electrons from donors, like reduced nicotinamide adenine dinucleotide (NADH) or quinols. On the extracytosolic side of the membrane,

**eLife digest** In bacteria, certain chemical reactions required for life do not take place directly inside the cells. For instance, 'redox' reactions essential to gather minerals, repair proteins and obtain energy are localised in the membranes and space that surround a bacterium. These chemical reactions involve electrons being transferred from one molecule to another in a cascade that connects the exterior of a cell to its internal space.

The enzyme ApbE allows proteins to perform electron transfer by equipping them with ring-like compounds called flavins, through a process known as flavinylation. Yet, the prevalence of flavinylation in bacteria and the scope of redox reactions it facilitates has remained unclear.

To investigate this question, Méheust, Huang et al. analysed over 30,000 bacterial genomes, finding genes essential for ApbE flavinylation in about half of all bacterial species across the tree of life. The role of ApbE-flavinylated proteins was then deciphered using a 'guilt by association' approach. In bacteria, genes that perform similar roles are often close to each other in the genome, which helps to infer the function of a protein coded by a specific gene. This approach revealed that flavinylation is involved in processes that allow bacteria to acquire iron and to use various energy sources. A number of interesting proteins were also identified, including a group that carry multiple flavins, and could therefore, in theory, transfer electrons over long distances. This discovery could be relevant to bioelectronic applications, which are already considering another class of bacterial electron-carrying molecules as candidates to form minuscule electric wires.

electron transfer between membrane and extracytosolic proteins is generally mediated by a redox-active protein, such as a cytochrome or thioredoxin-like protein. Cytochromes use redox-active hemes as cofactors and are often important for transferring electrons between membrane components and respiratory enzymes (*Bertini et al., 2006*). Thioredoxin-like proteins employ pairs of cysteines that cycle between redox states and are often involved in extracytosolic protein maturation and repair (*Cho and Collet, 2013*; *Collet and Messens, 2010*).

Flavins are a group of molecules that contain a conserved redox-active isoalloxazine ring system. In a reversible manner, the oxidized state of the flavin isoalloxazine ring system can undergo a one-electron reduction to generate a semiquinone state or a two-electron reduction to generate a hydroquinone state. Many microbes synthesize the flavin-derivative riboflavin (or vitamin B2), which can be phosphorylated to yield flavin mononucleotide (FMN) and further adenylated to flavin adenine dinucleotide (FAD). In part, because of their ability to transition between multiple redox states, FMN and FAD are well suited to act as enzyme cofactors. Proteins that bind flavins (flavoproteins) are common throughout nature and function in diverse redox activities (*Fraaije and Mattevi, 2000*).

In addition to the well-established electron transfer mediators, like cytochromes and thioredoxin-like proteins, an evolutionarily conserved FMN-binding domain has more recently been implicated in extracytosolic electron transfer (*Zhou et al., 1999*). FMN-binding domains are post-translationally flavinylated by the ApbE enzyme, which transfers the FMN portion of a substrate FAD molecule to a conserved [S/T]GA[S/T]-like sequence motif (flavinylated amino acid in bold) (*Figure 1A*; *Bertsova et al., 2013*). The resulting phosphoester bond irreversibly links the FMN ribitylphosphate group to a serine/threonine hydroxyl side chain in the FMN-binding domain.

While covalently bound flavins are a relatively common feature of flavoproteins, the ApbE-catalyzed reaction is unusual. In most flavinylated proteins, a covalent bond links the flavin's isoalloxazine ring to an amino acid side chain (*Macheroux et al., 2011*). This type of linkage induces a change in the flavin's redox potential that facilitates the protein's activity (*Macheroux et al., 2011*). By contrast, the ApbE-catalyzed phosphoester linkage is outside the flavin's isoalloxazine ring and unlikely to impact the cofactor's redox potential. Since non-flavinylated FMN-binding domains have low flavin-binding affinity, the ApbE-catalyzed reaction may simply function to secure the cofactor to the protein (*Borshchevskiy et al., 2015*). Once the flavin is linked to the FMN-binding domain, interactions with the folded protein stabilize the flavin's semiquinone state in a fashion that presumably enhances electron transfer (*Barquera, 2014*; *Backiel et al., 2008*).

ApbE-flavinylated FMN-binding domains have been found to be critical for five characterized extracytosolic electron transfer systems. These systems include the cation-pumping NADH:quinone

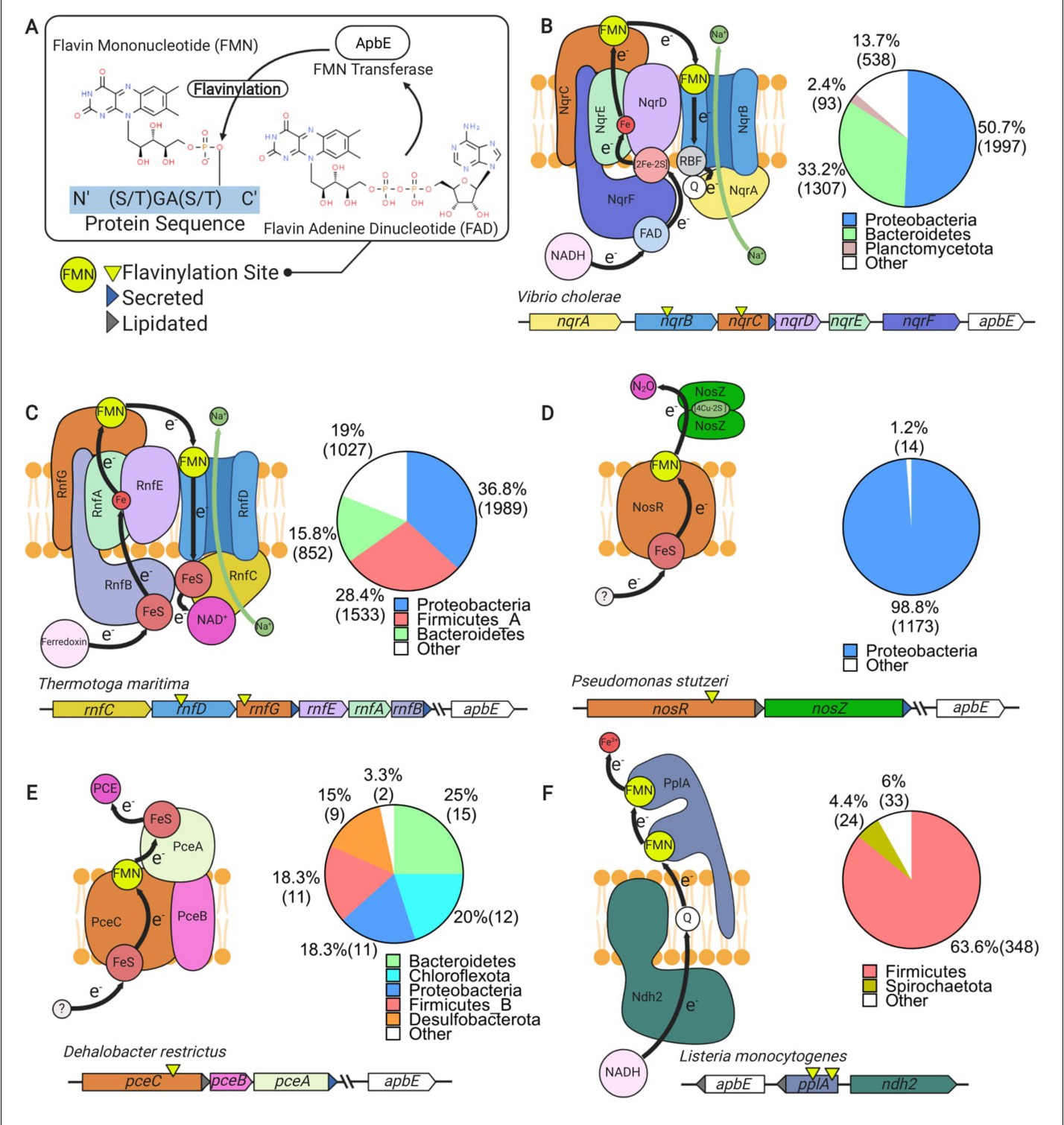

**Figure 1.** Characterized extracytosolic electron transfer systems with an ApbE-flavinylated subunit. (A) The post-translational flavinylation reaction catalyzed by the ApbE enzyme. (B–F) Models of previously characterized electron transfer systems that include an ApbE-flavinylated subunit. Gene clusters that encode each system are shown, with arrowheads indicating gene orientation and additional features shown in the key. 'Secreted' and 'lipidated' refer to the presence of computationally predicted signal peptides and lipidation sites, respectively. Black arrows trace proposed electron transfer pathways. Depicted systems and primary references for the model are: (B) NADH:quinone oxidoreductase (NQR) (*Steuber et al., 2014*), (C) *Rhodobacter* nitrogen fixation (RNF) (*Kuhns et al., 2020*), (D) nitrous oxide reduction (*Zhang et al., 2017*), (E) organohalide reduction (*Buttet et al.,*

*Figure 1 continued on next page*

*Figure 1 continued*

*2018*), and (F) extracellular electron transfer (*Light et al., 2018*). Accompanying pie charts show the number of genomes within each phylum that encode the electron transfer system.

The online version of this article includes the following figure supplement(s) for figure 1:

**Figure supplement 1.** Distribution of characterized extracytosolic electron transfer systems within prokaryotic genomes.

oxidoreductase (NQR) and *Rhodobacter* nitrogen fixation (RNF) complexes (*Figure 1B and C*), nitrous oxide and organohalide respiratory complexes (*Figure 1D and E*), and a Gram-positive extracellular electron transfer system (*Figure 1F*; *Backiel et al., 2008*; *Buttet et al., 2018*; *Light et al., 2018*; *Zhang et al., 2017*; *Zhou et al., 1999*). In addition, ApbE-flavinylated [S/T]GA[S/T]-like sequence motifs in homologous extracytosolic fumarate and urocanate reductases facilitate transfer from respiratory electron transport chains (*Bogachev et al., 2012*; *Kees et al., 2019*; *Light et al., 2019*). Notably, each of these characterized activities links AbpE flavinylation to a different aspect of microbial cellular respiration, with the NQR and RNF complexes being particularly noteworthy. These systems are widely distributed throughout microbial life and catalyze key intermediate steps in the energy metabolism of numerous microbes (*Barquera, 2014*; *Buckel and Thauer, 2018*; *Reyes-Prieto et al., 2014*).

In recent years, large-scale comparative genomic analyses have emerged as a powerful tool for discovering functionally and/or mechanistically related features of prokaryotic biology (*Burstein et al., 2017*; *Crits-Christoph et al., 2018*; *Doron et al., 2018*). Here, we develop a 'guilt by association' approach (*Aravind, 2000*) (summarized in *Figure 2—figure supplement 1*) that exploits genomic diversity to contextualize the significance of extracytosolic flavinylation. Our analysis of flavinylation-associated gene clusters provides evidence of widespread flavinylation throughout bacteria and uncovers new connections to respiration and iron assimilation. We further identify uncharacterized aspects of extracytosolic flavinylation, including novel ApbE substrates and a class of multi-flavinylated proteins. These findings place ApbE-flavinylated proteins alongside cytochromes and thioredoxin-like proteins as central mediators of bacterial extracytosolic electron transfer.

## Results

### Evidence of widespread extracytosolic flavinylation within bacteria

As all previously characterized flavinylation systems contain genes that encode for an ApbE enzyme and a substrate that contains an FMN-binding domain, we reasoned that these features are indicative of flavinylation-mediated electron transfer. To identify flavinylation-mediated electron transfer systems, we searched for genes with ApbE (Pfam accession PF02424) or FMN-binding domains (Pfam accession PF04205) within a collection of 31,910 genomes that are representative of the genetic diversity of the prokaryotes (*Parks et al., 2018*; *Parks et al., 2020*). We found that 18,965 of bacterial genomes and 238 of archaeal genomes encode an FMN-binding domain-containing protein and/or an ApbE enzyme. To focus the search on extracytosolic electron transfer, we eliminated genes that lacked a computationally predicted signal peptide or lipidation site. This analysis provides evidence that ~50% of the bacterial (15,095) and ~4% of archaeal (63) genomes possess extracytosolic flavinylation components (*Supplementary file 1*). A phylogenetic analysis of the resulting dataset reveals that extracytosolic flavinylation components are broadly distributed across bacterial life – though strikingly underrepresented in Cyanobacteria and the candidate phyla radiation (*Figure 2*).

We next took advantage of the colocalization of genes for multi-subunit complexes on the genome to determine the prevalence and phylogenetic distribution of previously characterized flavinylated systems. We devised operational definitions in which the close proximity of a key gene in each system to an FMN-binding- or ApbE-containing gene was used to assign clusters to characterized systems (*Supplementary file 2*). These analyses revealed characterized systems in 8928 genomes broadly distributed across bacterial life (*Figure 2*).

In addition to assigning extracytosolic electron transfer functions, our analyses identified 6230 genomes that contained evidence of extracytosolic flavinylation but lacked a characterized extracytosolic electron transfer system. We also found that many genomes have multiple ApbE and/or

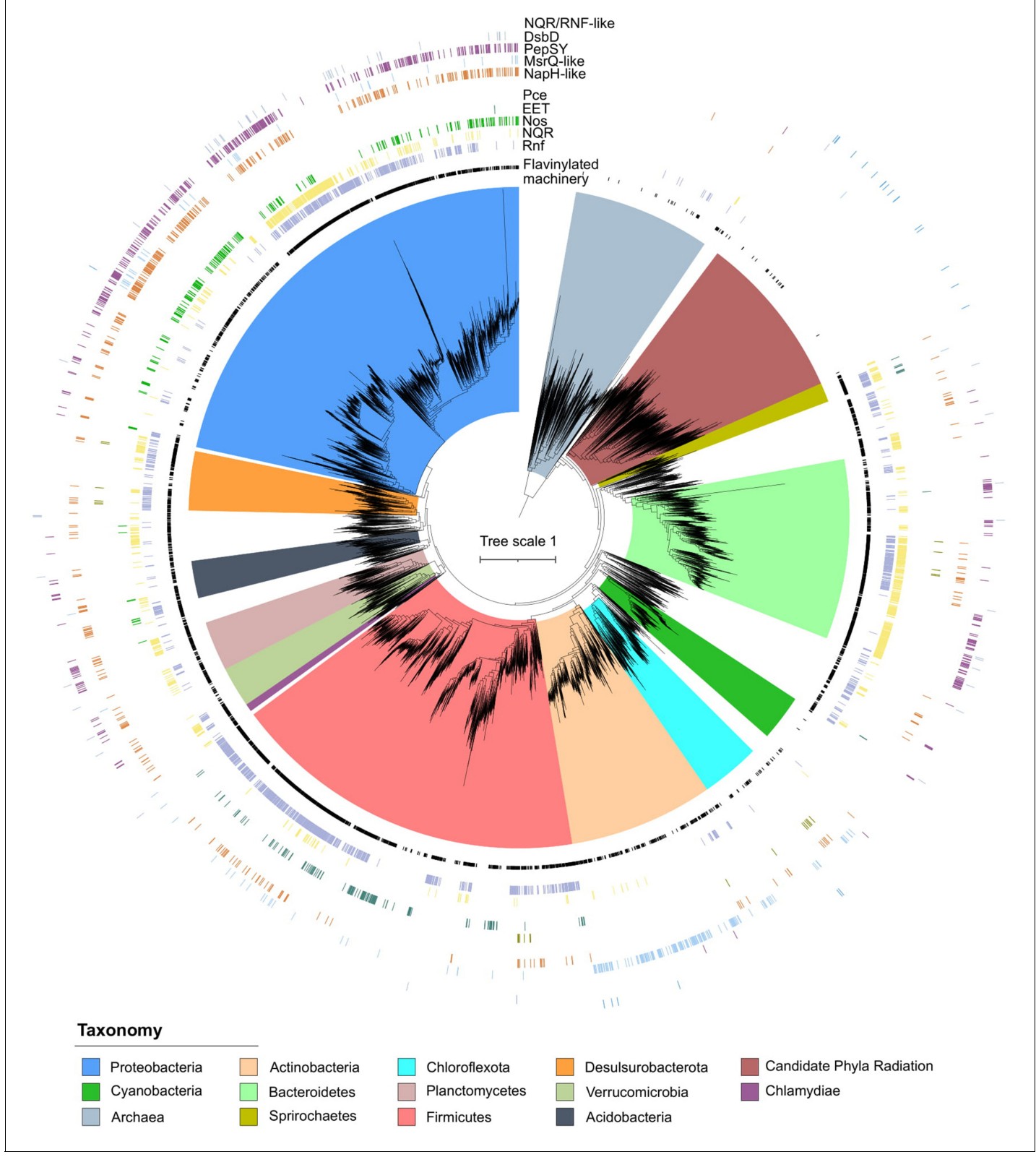

**Figure 2.** Phylogenetic distribution of extracytosolic protein flavinylation in prokaryotes. Phylogenetic reconstruction of the evolutionary history of 9152 genomes representing 97% of the diversity available at the genus level in the GTDB (9428 distinct genera) (*Parks et al., 2020*). The maximum likelihood tree was constructed based on a concatenated alignment of 14 ribosomal proteins under an LG + I + G4 model of evolution (2850 amino acid sites). The inner to outer rings display the presence in one genome of the genus of extracytosolic flavinylated proteins (i.e., genomes encoding a signal

*Figure 2 continued on next page*

*Figure 2 continued*

peptide-containing flavin mononucleotide [FMN]-binding or ApbE) (flavinylation machinery), the presence of a *Rhodobacter* nitrogen fixation (RNF) system, an NADH:quinone oxidoreductase (NQR) system, a Nos system, an EET system, and an organohalide reduction system (Pce), the presence of an NapH-like system, an MsrQ-like system, a PepSY system, a DsbD system, and an NQR/RNF-like system. The scale bar indicates the mean number of substitutions per site.

The online version of this article includes the following figure supplement(s) for figure 2:

**Figure supplement 1.** Summary of the discovery strategy followed by described studies.

FMN-binding genes and thus likely possess multiple extracytosolic electron transfer functions (*Figure 1—figure supplement 1*). As these observations suggested that a significant proportion of extracytosolic electron transfer systems remain uncharacterized, we next turned to the identification of uncategorized flavinylation-associated gene clusters.

## DUF2271 and DUF3570 domains of unknown function are ApbE substrates

Our initial approach only considered FMN-binding domains as potential flavinylation substrates. However, preliminary analyses identified 17% of genomes (2571 total) within our dataset that encoded an ApbE but no FMN-binding domain-containing protein – implying that some ApbE substrates lack an FMN-binding domain (*Figure 3—figure supplement 1*). To identify novel ApbE protein substrates, we examined the genomic context of these 'orphan' *apbE* genes, looking for gene cluster patterns conserved across multiple genomes (*Supplementary file 1*).

We observed that a subset of orphan *apbE* genes are associated with DUF2271 – an ~135 amino acid *d*omain of *u*nknown *f*unction. In some genomes a single gene encodes a protein with both ApbE and DUF2271 domains (e.g., NCBI accession RUL87804.1), but more commonly a separate DUF2271 gene is part of a gene cluster that includes an *apbE*. Consistent with DUF2271 serving as a flavinylation substrate, we identified a conserved [S/T]GA[S/T] motif within ApbE-associated DUF2271s (*Figure 3A*). We expressed the *Amantichitinum ursilacus* DUF2271 protein in *Escherichia coli* and found that it was flavinylated in the presence of its cognate ApbE (*Figure 3B*). Consistent with the conserved [S/T]GA[S/T] motif representing the sole ApbE target, we found that replacing the threonine at the predicted flavinylation site with an alanine abrogated flavinylation (*Figure 3B*).

We observed another subset of orphan *apbE* genes that are part of gene clusters that contain a gene annotated as DUF3570 – an ~420 amino acid domain of unknown function. Consistent with DUF3570 serving as a flavinylation substrate, we identified two conserved [S/T]GA[S/T] motifs within ApbE-associated DUF3570s (*Figure 3C*). Using the coexpression approach described above, we confirmed that the *Chlorobium luteolum* DUF3570 was flavinylated in the presence of its cognate ApbE (*Figure 3D*). Consistent with both of the identified [S/T]GA[S/T] motifs being modified, we found that replacing serines with alanines at both predicted flavinylation sites was required to abrogate flavinylation (*Figure 3D*).

These results suggest that DUF2271 and DUF3570 are novel ApbE substrates. Including DUF2271 and DUF3570 within our analyses significantly expanded the number and diversity of predicted ApbE substrates and decreased the number of orphan *apbE* genomes within the dataset from 17% to 4% (670 total) (*Figure 3—figure supplement 1*). Subsequent analyses thus likely account for a significant fraction of flavinylation substrates – though the fact that orphan *apbE* genomes remain suggests that less prevalent ApbE substrates remain unidentified.

## Identification of flavinylation-associated transmembrane electron transfer components

We next sought to gain insight into novel roles of protein flavinylation. Characterized flavinylation-based electron transfer systems are minimally defined by a cytosolic electron donor, a transmembrane electron transfer apparatus, and the flavinylated extracytosolic electron acceptor – though, in principle, the direction of electron flow could be reversed. We reasoned that identification of cytosolic and membrane components was important for understanding the extracytosolic capability of uncharacterized flavinylated systems.

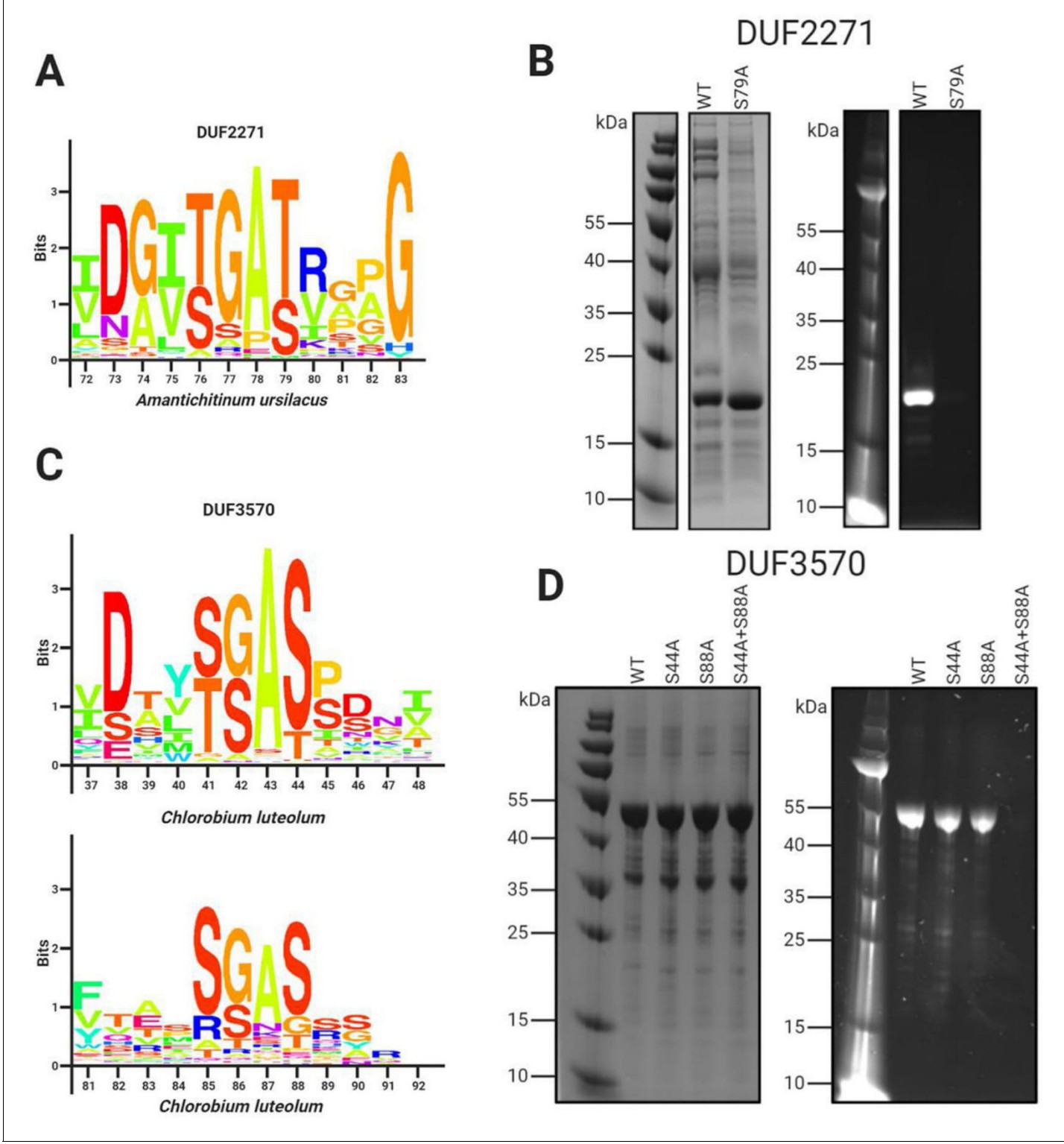

**Figure 3.** Domains of unknown function DUF2271 and DUF3570 are flavinylated by ApbE. (**A**) Conserved sequence motif within 282 flavinylation-associated DUF2271 proteins. Letter size (bits) is proportional to amino acid frequency at each position in the sequence alignment. Amino acid numbering corresponds to the *Amantichitinum ursilacus* DUF2271 sequence. (**B**) SDS-PAGE of purified *A. ursilacus* DUF2271 variants coexpressed in *Escherichia coli* with their cognate ApbE. The gel is shown with coomassie stain (left) and under ultraviolet illumination (right). (**C**) Conserved sequence motif within 228 flavinylation-associated DUF3570 proteins. Letter size (bits) is proportional to amino acid frequency at each position in the sequence

*Figure 3 continued on next page*

*Figure 3 continued*

alignment. Amino acid numbering corresponds to the *Chlorobium luteolum* DUF3570 sequence. (D) SDS-PAGE of purified *C. luteolum* DUF3570 variants coexpressed in *E. coli* with their cognate ApbE. The gel is shown with coomassie stain (left) and under ultraviolet illumination (right).

The online version of this article includes the following figure supplement(s) for figure 3:

**Figure supplement 1.** Presence of *apbE* and its putative substrates in prokaryotic genomes.

To clarify the role of flavinylation-associated gene clusters, we analyzed the genomic context of a subset of representative genes with DUF3570, DUF2271, FMN-binding, and ApbE domains that were not assigned to a characterized system in our initial analyses. Annotations for the five upstream and five downstream genes were manually reviewed. From these gene clusters, we identified five putative transmembrane electron transfer apparatuses that are present in 6183 genomes, including 3635 of the 6230 genomes that lack a characterized extracytosolic electron transfer system (*Figure 4*, *Figure 4—figure supplement 1* and *Supplementary file 2*). The following subsections describe the organization and likely functions of these systems. Annotations of the gene clusters associated with the flavinylated systems are provided in *Supplementary file 1*.

## NapH-like systems

We identified 2465 flavinylation gene clusters in 2153 genomes that encode an 'NapH-like' iron-sulfur cluster-binding protein (Pfam accession PF12801 or KEGG accession K19339) that contains several transmembrane helices (*Figure 2* and *Supplementary file 1*). These proteins are homologous to NapH, the putative quinone-binding subunit of periplasmic nitrate reductase, and exhibit a broad phylogenetic distribution (*Figure 4A*; *Brondijk et al., 2002*). The majority of NapH-like proteins identified in our analysis contain an extracytosolic N-terminal FMN-binding domain (1926 gene clusters) whereas the remaining ones (539 gene clusters) are in a genomic locus with a second gene that encodes an FMN-binding domain-containing protein. These proteins likely receive electrons from a donor (probably a quinol in some cases) that are transferred via the iron-sulfur cluster across the membrane (*Figure 4A*).

Several lines of evidence suggest that many NapH-like proteins function with respiratory oxidoreductases. Previously characterized NapH-like proteins PceC and NosZ have been shown to be flavinylated and are part of gene clusters for organohalide and nitrous oxide reduction, respectively (*Figure 1*; *Buttet et al., 2018*; *Zhang et al., 2017*). These previously characterized NapH-like systems were identified within 1197 genomes within our dataset (*Figure 2*). We also identified NapH-like gene clusters with an extracytosolic nitrite reductase in 133 genomes or an ethanol oxidase in 172 genomes. Thus, while the reactions catalyzed by the majority of NapH-like systems remain unknown, this electron transfer apparatus seems to be modularly employed to facilitate electron transfer to reductases or from oxidases (*Figure 4A*). This modularity of NapH-oxidoreductase associations is further underscored by a phylogenetic analysis, which suggests that NapH-like associations with *nirS* and *exaA* evolved independently multiple times (*Figure 4—figure supplement 2*).

## MsrQ-like systems

We identified 1797 flavinylation gene clusters in 1468 genomes that encode an 'MsrQ-like' (Pfam accession PF01794) protein (*Supplementary file 1*). These clusters are broadly conserved in Actinobacteria and are infrequently identified in other lineages (*Figure 2* and *Figure 4B*). MsrQ-like proteins are predicted to have six transmembrane helices and two heme-binding sites. MsrQ-like proteins are homologous to MsrQ, the quinone-binding subunit of periplasmic methionine sulfoxide reductase (*Gennaris et al., 2015*). MsrQ-like proteins are also distantly related to eukaryotic proteins that function in transmembrane electron transfer, including NADPH oxidase and STEAP iron reductases (*Zhang et al., 2013*).

MsrQ-like gene clusters typically include *apbE* and an FMN-binding domain-containing gene. MsrQ-like proteins often contain a C-terminal NAD-binding domain (Pfam accession PF00175, 1437 gene clusters). In other cases, the MsrQ-like gene clusters include an NuoF-like protein (Pfam accession PF10589 , 31 gene clusters), homologous to the NAD-binding subunit of complex I. At least a subset of MsrQ-like proteins thus likely use NAD(P)H as a cytosolic electron donor (*Figure 4B*). We also identified 153 MsrQ-like gene clusters that encode a protein homologous to the eukaryotic

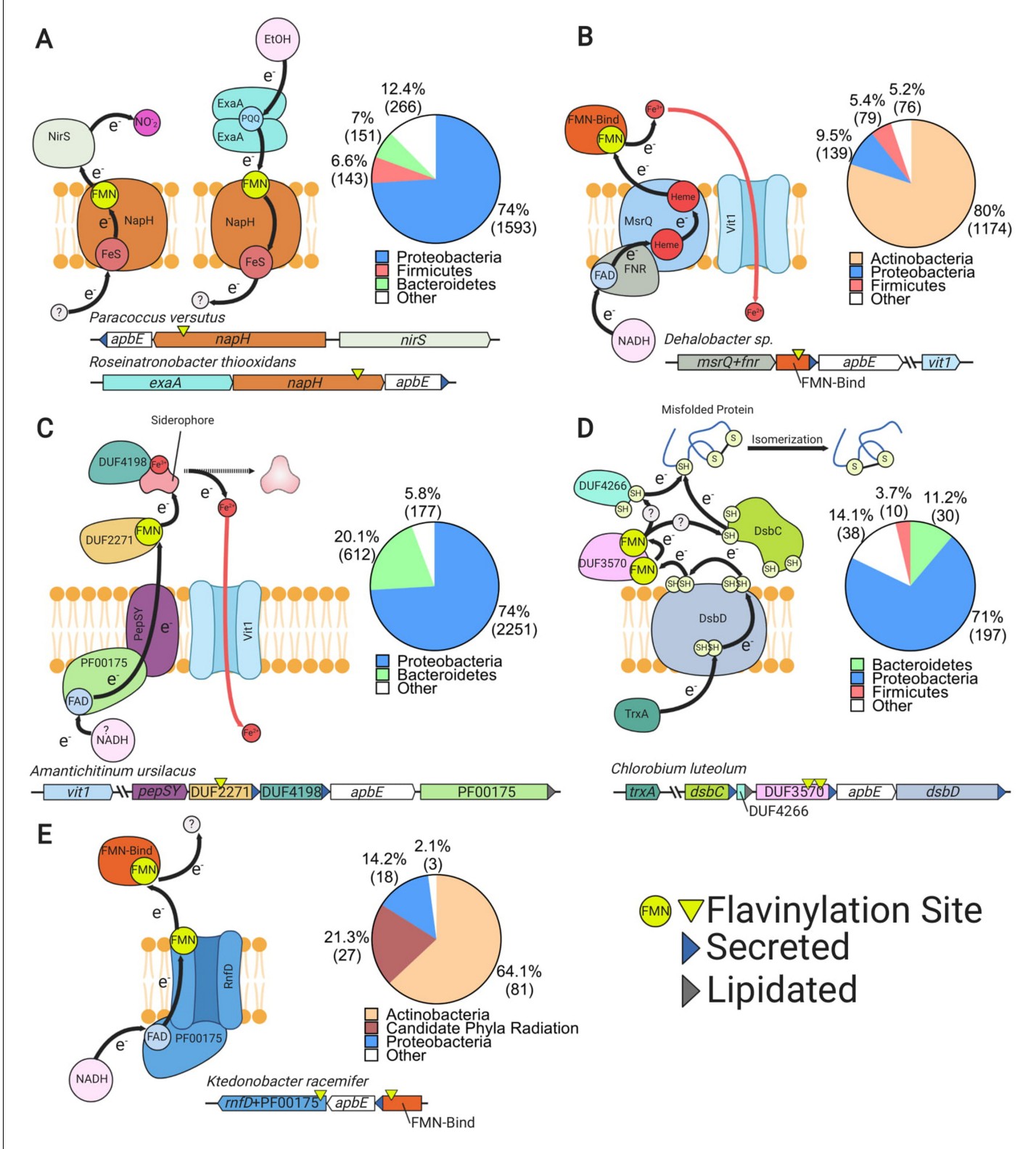

**Figure 4.** Uncharacterized extracytosolic electron transfer systems with an ApbE-flavinylated subunit. Representative gene clusters and hypothesized models of the electron transfer systems they encode: (A) NapH-like (*Moreno-Vivián et al., 1999*), (B) MsrQ-like (*Juillan-Binard et al., 2017*), (C) PepSY (*Yeats et al., 2004*), (D) DsbD (*Bushweller, 2020*), and (E) RNF/NQR-like (*Steuber et al., 2014*). System names are based on sequence homology of a membrane protein encoded within the cluster to the referenced protein (see main text for additional context). Black arrows trace proposed electron
*Figure 4 continued on next page*

*Figure 4 continued*

transfer pathways. Accompanying pie charts show the number of genomes within each phylum that encode the electron transfer system. 'Secreted' and 'lipidated' refer to the presence of computationally predicted signal peptides and lipidation sites, respectively.

The online version of this article includes the following figure supplement(s) for figure 4:

**Figure supplement 1.** Distribution of uncharacterized extracytosolic electron transfer systems within prokaryotic genomes.

**Figure supplement 2.** Maximum likelihood phylogeny of flavinylation-associated NapH-like proteins.

ferrous iron transporter VIT1 (Pfam accession PF01988). This association suggests that some MsrQ-like systems function as assimilatory iron reductases that facilitate iron uptake through VIT1 (*Figure 4B*).

## PepSY-like systems

We identified 3220 flavinylation gene clusters in 3040 genomes that encode a PepSY-like (Pfam accessions PF03929 and PF16357) protein (*Supplementary file 1*). PepSY-like proteins contain three transmembrane helices and are broadly distributed throughout Gram-negative bacteria (*Figures 2* and *4C*). PepSY-like gene clusters frequently encode an ApbE enzyme, the flavinylation substrate DUF2271 (Pfam accession PF10029, 2833 clusters), and a secreted DUF4198 (Pfam accession PF10670, 1500 clusters) protein. Little is known about the structure or function of PepSY-like proteins. The only characterized PepSY-like homolog, *Vibrio cholerae* VciB (which is not associated with identified flavinylation components), was reported to possess extracytosolic iron reductase activity (*Peng and Payne, 2017*) an — an activity consistent with transmembrane electron transfer activity. We further identified 1077 PepSY-like genes that have a NAD-binding domain (Pfam accession PF00175), suggesting that a subset of these proteins use cytosolic NADH as an electron donor (*Figure 4C*).

Several observations implicate a role for identified PepSY-like gene clusters in iron reduction and assimilation. First, a functional connection to MsrQ-like proteins is suggested by our observation that 108 gene clusters contain both PepSY-like and MsrQ-like genes. Second, 167 PepSY-like gene clusters contain a VIT1 (Pfam accession PF01988) ferrous iron transporter (*Figure 4C*). Finally, PepSY-like gene clusters have been shown to be repressed by Fur (the primary transcription regulator that responds to iron limitation) in *Shewanella oneidensis* and *Caulobacter crescentus* (*da Silva Neto et al., 2013*; *Wan et al., 2004*). Moreover, in some cases, PepSY-like proteins may be actively involved in the extraction of siderophore-bound iron, as periplasmic reduction has been shown to be important for the uptake of siderophore-bound iron and 231 PepSY-like gene clusters encode a TonB receptor (Pfam accession PF03544) related to well-characterized outer membrane siderophore transporters (*Figure 4C*; *Liu et al., 2018a*; *Manck et al., 2020*).

## DsbD systems

We identified 285 flavinylation gene clusters in 275 genomes that contain a DsbD protein (Pfam accession PF02683) (*Figure 2* and *Supplementary file 1*). DsbD is part of a well-studied transmembrane protein family that uses cysteine pairs to transfer electrons across the membrane (*Krupp et al., 2001*; *Missiakas et al., 1995*). DsbD family proteins generally transfer electrons onto extracytosolic thioredoxin-like proteins, which in turn use a similar thiol-disulfide exchange chemistry to promote extracytosolic redox-dependent activities, such as oxidative protein folding (*Cho and Collet, 2013*).

The DsbD gene clusters identified in our analyses typically include genes for ApbE, the flavinylation substrate DUF3570 (Pfam accession PF12094, 227 clusters), a thioredoxin-like protein (Pfam accession PF13899, 183 clusters), and a DUF4266 protein (Pfam accession PF14086, 221 clusters). DUF4266 is a small secreted protein with a highly conserved C-terminal cysteine – (any amino acid) – cysteine (CXC) sequence motif that may undergo redox cycling.

We also detected 316 gene clusters that encode ApbE, DUF3570, thioredoxin-like, and DUF4266 but do not colocalize on the genome with DsbD – implying that this system may be more common than is revealed by the DsbD-dependent analysis presented in *Figure 2*. These observations suggest that a hybrid thioredoxin-like/flavinylation-based system receives electrons from DsbD in some bacteria (*Figure 4D*). While the function of DUF3570-based electron transfer remains uncertain, 25 gene

clusters contain a VIT1 ferrous iron transporter and thus may play a role in iron assimilation (*Supplementary file 1*).

## NQR/RNF-like systems

We identified 127 flavinylation gene clusters in 127 genomes, primarily from the Actinobacteria and candidate phyla radiation, with evidence of partial NQR or RNF systems (*Figure 2*). As shown in *Figure 1*, the NQR and RNF complexes contain a common core apparatus with two transmembrane electron transfer pathways that together achieve a semicircular electron flow (*Juárez et al., 2010*; *Steuber et al., 2014*). A first path takes electrons from the cytosol to the extracytosolic FMN-binding domain, while a second path takes electrons from the FMN-binding domain back to a cytosolic substrate. NQR/RNF-like gene clusters encode for components associated with a single electron transfer pathway and thus likely function for unidirectional electron flow (*Figure 4E*).

NQR/RNF-like gene clusters encode a protein with an N-terminal membrane domain that is homologous to NqrB/RnfD and a cytosolic C-terminal domain homologous to the NAD-binding domain NqrF (Pfam accession PF00175). NqrB/RnfD is flavinylated by ApbE and is thought to play a role in electron transfer across the membrane (*Juárez et al., 2010*). This system presumably uses NAD(P)H as a cytosolic electron donor for electron transfer to an extracytosolic FMN-binding domain. Six NQR/RNF-like gene clusters contain a VIT1 ferrous iron transporter, suggesting that some of these systems function in iron assimilation (*Supplementary file 1*).

## Other flavinylated proteins implicated in iron assimilation and respiration

We next asked about the function of flavinylation-associated gene clusters that lack a core transmembrane electron transfer system. This category of extracytosolic proteins presumably relies on proteins encoded elsewhere on the genome to link up with membrane electron pools. Inspection of these gene clusters led to the identification of two noteworthy examples that are described in the following subsections.

## P19-associated iron assimilation

A gene cluster that includes the ferrous iron-binding protein P19 (Pfam accession PF10634) and the FTR1 (Pfam accession PF03239) iron transporter has previously been shown to encode a mechanistically uncharacterized system involved in iron assimilation (*Chan et al., 2010*). We identified 260 P19 gene clusters, mostly in Gram-positive bacteria, that contain a gene with an FMN-binding domain (*Figure 5—figure supplement 1A and B* and *Supplementary file 1*). Interestingly, many Gram-negative P19 gene clusters lack an FMN-binding gene but contain an additional thioredoxin-like gene (*Liu et al., 2018b*; *Figure 5—figure supplement 1B*). The role of the FMN-binding/thioredoxin-like protein in these systems has not been defined but, similar to other assimilatory iron reductases, could engage in redox chemistry to facilitate iron uptake. These observations thus establish another connection between flavinylation and microbial iron assimilation and highlight functional parallels between flavinylation and thioredoxin-like extracytosolic electron transfer.

## Fumarate reductase-like oxidoreductases

Fumarate reductase-like enzymes (members of the Pfam accession PF00890 enzyme superfamily) are a group of evolutionarily related proteins that catalyze a variety of redox reactions – though phylogenetic analyses suggest that substrates for many members of the superfamily remain unknown (*Jardim-Messeder et al., 2017*; *Light et al., 2019*). We observe that fumarate reductase-like enzymes often contain an FMN-binding domain (3070 proteins encoded by 2979 distinct gene clusters in 1236 genomes) or are part of gene clusters (±2 genes) that contain an FMN-binding domain (189 gene clusters in 171 genomes) (*Supplementary file 1*). Several characterized fumarate reductase-like enzymes (including fumarate, urocanate, and methacrylate reductases) are extracytosolic and function in respiration (*Bogachev et al., 2012*; *Light et al., 2019*; *Mikoulinskaia et al., 1999*). The *Listeria monocytogenes* fumarate reductase and the *S. oneidensis* urocanate reductase have been shown to be flavinylated (*Bogachev et al., 2012*; *Light et al., 2019*). In both cases, the flavinylation motif is thought to facilitate electron transfer from electron transport chain components encoded elsewhere on the genome to the enzyme active site (*Kees et al., 2019*; *Light et al., 2019*). These

observations thus suggest that FMN-binding domains mediate electron transfer from membrane components to a prevalent class of extracytosolic reductases and highlight another connection between flavinylation and respiration.

## Modular flavinylation/cytochrome usage in respiratory systems

A comparison of extracytosolic electron transfer systems identified in our analyses revealed multiple instances in which cytochrome and flavinylated electron transfer components appear to be performing similar electron transfer roles within related systems. For example, we identified a flavinylated NapH-like protein that is well situated to mediate electron transfer from the extracytosolic alcohol oxidase to the electron transport chain, where a cytochrome c protein has been shown to play this role in other microbes (*Figure 5A*; *Schobert and Görisch, 1999*).

Another example of this dynamic is provided by a comparison of fumarate reductases. The extracytosolic *S. oneidensis* fumarate reductase contains an N-terminal multi-heme cytochrome c domain (Pfam accession PF14537), whereas the related *L. monocytogenes* enzyme uses an FMN-binding domain to connect to the electron transport chain (*Figure 5B*; *DiChristina and DeLong, 1994*; *Light et al., 2019*). Different microbes thus seem to utilize flavinylation and cytochrome domains in a similar fashion to link respiratory enzymes to electron transport chains.

We further found the type of flavinylation/cytochrome substitution observed for fumarate reductase to be indicative of a broader pattern within fumarate reductase-like enzymes. We identified 147 gene clusters within 108 genomes that encode separate fumarate reductase-like and cytochrome proteins and 879 genes within 360 genomes that encode a single protein with both fumarate reductase-like and cytochrome domains (*Supplementary file 3*). Many genomes encode multiple fumarate reductase-like paralogs and 99 genomes encode both cytochrome- and flavinylation-associated enzymes (*Supplementary file 3*). This dynamic is exemplified by *S. oneidensis*, which in addition to the mentioned cytochrome-associated fumarate reductase contains a flavinylated urocanate reductase that also exhibits a respiratory function (*Figure 5C*; *Bogachev et al., 2012*). Broadly similar flavinylation- and cytochrome-based extracytosolic electron transfer mechanisms thus seemingly coexist within some microbes.

## Multi-flavinylated proteins may facilitate longer distance electron transfer

Multi-heme cytochromes are proteins that bind multiple hemes to achieve longer distance extracytosolic electron transfer (*Blumberger, 2018*). Among other functions, multi-heme proteins are important for transferring electrons across the cell envelope to insoluble electron acceptors that are inaccessible within the cytosolic membrane (*El-Naggar et al., 2010*; *Wang et al., 2019*). We have observed that extracytosolic proteins with multiple FMN-binding domains are also common (2081 proteins in 1530 genomes), particularly in Gram-positive bacteria, and contain as many as 13 predicted flavinylation sites (*Figure 6A and B* and *Supplementary file 1*). Multi-cofactor binding properties thus establish another parallel between cytochrome and flavinylation-based electron transfer.

Gene cluster analyses provide some insight into the basis of multi-flavinylated protein electron transfer. We find that multi-flavinylated proteins are often associated with established transmembrane electron transfer components and thus likely receive electrons through conventional mechanisms (*Figure 6C*). We observe that some clusters contain large unannotated proteins with putative cell wall-binding domains – such as SLH (Pfam accession PF00395), Rib (Pfam accession PF08428), FIVAR (Pfam accession PF07554), or LysM (Pfam accession PF01476) – and are thus likely involved in redox chemistry within the cell wall or at the cell surface (*Figure 6D*). We also identified 157 clusters that encode a multi-flavinylated protein and additional proteins with FMN-binding domains (*Figure 6D and E*). These clusters encode as many as five proteins with FMN-binding domains and frequently contain multiple multi-flavinylated proteins (*Figure 6D*). These observations suggest that some multi-flavinylated proteins are part of multi-step electron transfer pathways and may form elaborate multi-subunit complexes that span the cell wall.

The role of multi-flavinylated gene clusters is generally unclear, with only a minority of clusters providing limited functional clues. We identified a subset of clusters that encode proteins with multiple fumarate reductase-like domains that likely establish an unusual multi-functional reductase platform (*Figure 6D and G*). We also identified a number of RNF clusters that contain multi-flavinylated

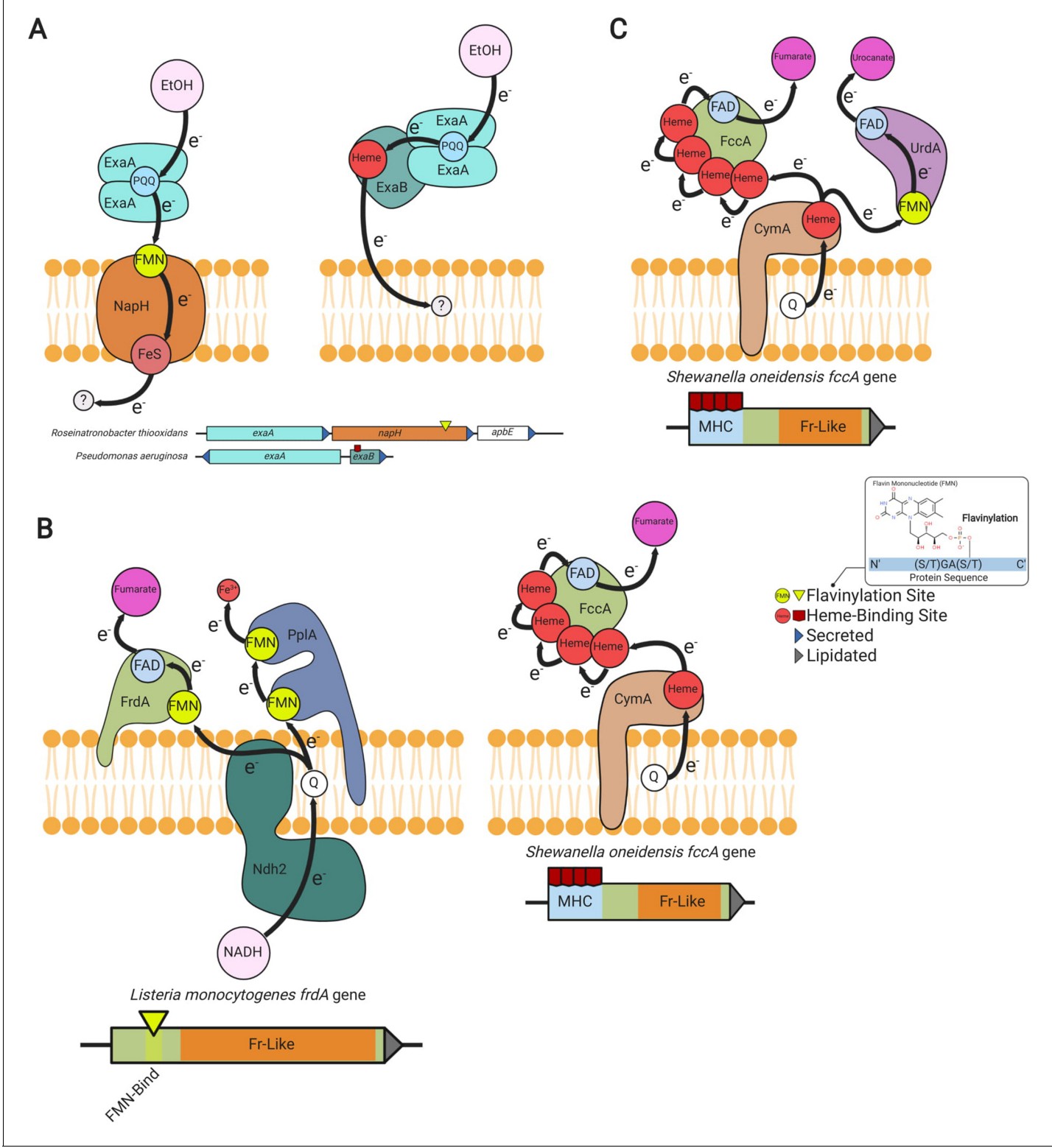

**Figure 5.** Modular flavinylation/cytochrome usage in extracytosolic electron transfer. (**A**) Model of the characterized *Pseudomonas aeruginosa* (*Görisch, 2003*; *Matsutani and Yakushi, 2018*) and uncharacterized *Rhiobacillus thiooxidans exaA* alcohol oxidase gene clusters and the proposed electron transfer pathways they encode. (**B**) Domain structure of *Listeria* monocytogenes and *Shewanella* oneidensis fumarate reductases (FrdA and FccA, respectively) and a model of their characterized interactions with electron transfer pathways (*Light et al., 2019*; *Kees et al., 2019*). (**C**)

*Figure 5 continued on next page*

Figure 5 continued

Characterized electron transfer pathways for the *S. oneidensis* fumarate reductase-like enzymes, fumarate reductase (FccA), and urocanate reductase (UrdA) (*Kees et al., 2019*).

The online version of this article includes the following figure supplement(s) for figure 5:

**Figure supplement 1.** Flavinylation and thioredoxin-like proteins encoded in P19 gene clusters.

RnfGs (the extracytosolic flavinylated subunit in RNF complexes) (*Figure 6C*). These multi-flavinylated RNF complexes likely provide a second electron transfer pathway that facilitates transfer to alternative extracytosolic acceptors. This type of bifurcated electron transfer would be similar to a multi-heme cytochrome-based transfer mechanism recently suggested in studies of a methanogen RNF (*Holmes et al., 2019*). Interestingly, we also identified a subset of multi-flavinylated RnfGs in the family Christensenellales that contain a multi-heme cytochrome domain and thus may assume additional functions relevant for cytochrome-based electron transfer (*Figure 6H*). While much remains to be learned, these preliminary observations are consistent with multi-flavinylated proteins establishing elaborate and functionally diverse electron transfer pathways.

## Discussion

Comparative analysis of gene clusters within collections of prokaryotic genomes has emerged as a powerful discovery tool in recent years. Our large-scale survey of diverse genomes extends this approach and reveals that ApbE-mediated flavinylation is a prominent feature of bacterial physiology. More granular analyses of gene clusters provide evidence consistent with extracytosolic flavinylation usage in diverse redox activities and suggest that modular properties facilitate the integration of flavinylated components with various biochemical processes. While additional genetic and/or biochemical studies will be required to develop a better understanding of the physiological roles of flavinylation, these preliminary observations are consistent with flavinylated proteins being important components of microbial extracytosolic electron transfer. Our findings suggest that, alongside thioredoxin-like proteins and cytochromes, ApbE-flavinylated proteins represent a third major class of mediators of extracytosolic electron transfer.

The existence of three mechanistically distinctive protein classes with some apparent functional interchangeability (*Figure 5*) stimulates fundamental questions about the environmental or physiological context that favors each system. Unfortunately, the multiple functions and widespread distribution of flavinylated proteins across diverse microbes make it difficult to identify unique features that distinguish microbes that encode flavinylation components. Nevertheless, the relationship between flavinylation- and cytochrome-based electron transfer is interesting. The apparent functional interchangeability in the electron transfer capabilities of iron- and flavin-containing proteins is reminiscent of the relationship between ferredoxins and flavodoxins. Ferredoxins and flavodoxins are redox-active proteins that function in a number of cytosolic redox activities and contain iron and flavin cofactors, respectively (*Yoch and Valentine, 1972*). Microbes switch from ferredoxin to flavodoxin usage in iron-poor environments – presumably because minimizing the demand for iron cofactors is important for conserving the cellular iron reserve in this context (*Knight et al., 1966*; *Smillie, 1965*). Based on this precedent, it seems plausible that flavinylation-based electron transfer mechanisms might be particularly advantageous within iron-poor environments. The coexistence of functionally similar but mechanistically distinctive flavinylation and cytochrome electron transfer components may thus, in part, reflect divergent resource management strategies in distinct environmental contexts.

Of the proteins linked to extracytosolic electron transfer by our studies, the class of multi-flavinylated proteins are particularly intriguing. These proteins may resemble multi-heme cytochromes in their use of multiple redox-active cofactors to achieve longer distance electron transfer. A particularly noteworthy aspect of multi-heme cytochromes concerns their ability to establish 'nanowires' that have a variety of potential biotechnological applications (*Blumberger, 2018*; *Liu et al., 2020*). Considering the unique redox properties of flavins, the electron-transferring behavior of multi-flavinylated proteins may provide an interesting juxtaposition to multi-heme cytochromes with implications for the development of novel redox-based biotechnologies. The function and mechanism of

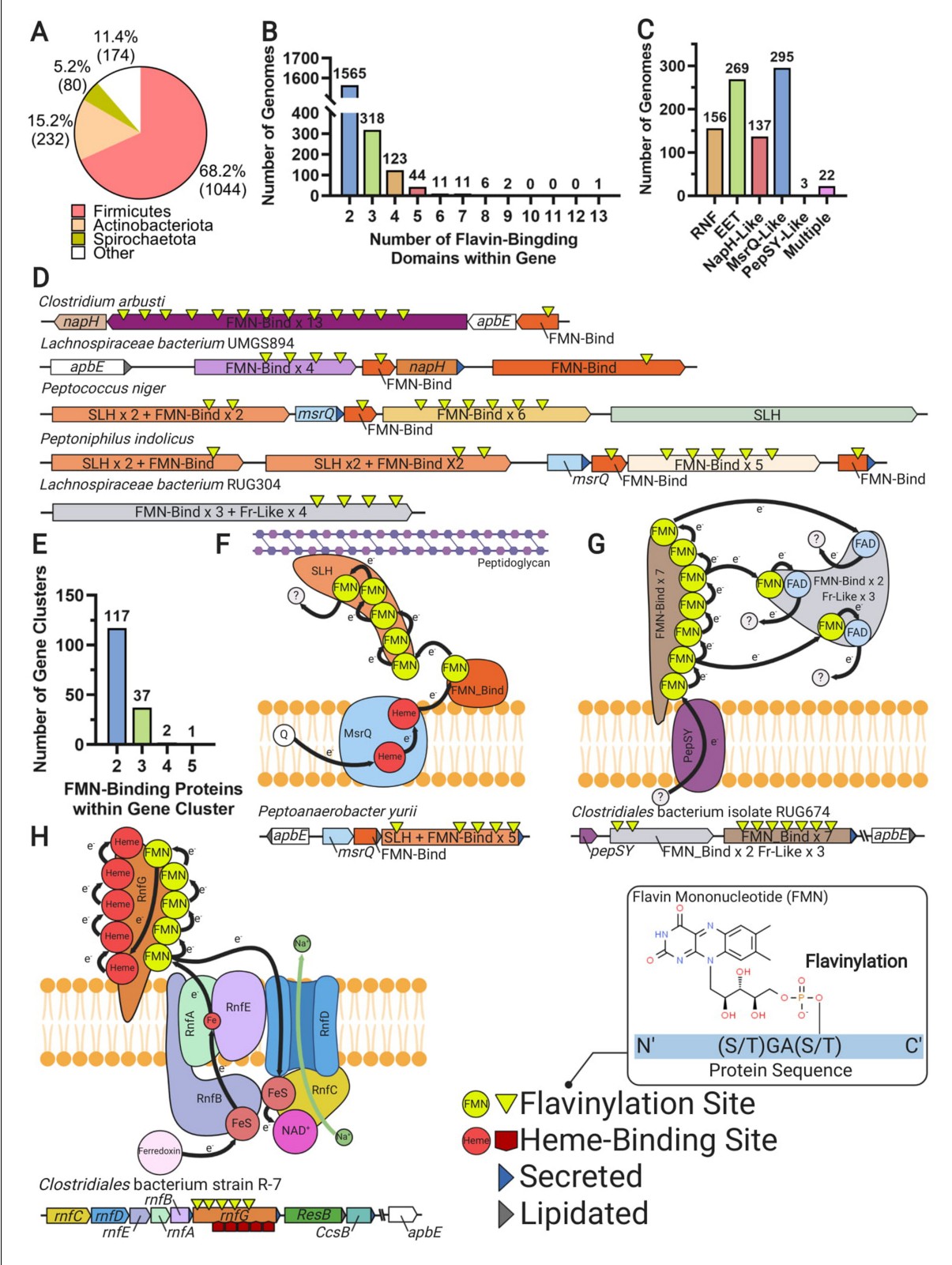

**Figure 6.** Multi-flavinylated proteins may possess novel electron transfer properties. (A) Pie chart showing the number of genomes within each phylum that contain a multi-flavinylated (i.e., >1 FMN-binding domain) protein. (B) Histogram showing the number of FMN-binding domains within identified multi-flavinylated proteins. (C) Histogram showing the number of multi-flavinylated protein gene clusters that encode one of the transmembrane electron transfer systems described in *Figures 1* and *4*. (D) Examples of multi-flavinylated protein gene clusters. Fumarate reductase-like domains are

*Figure 6 continued on next page*

*Figure 6 continued*

abbreviated as Fr-like. (E) Histogram showing the number of proteins containing at least one FMN-binding domain within multi-flavinylated protein gene clusters. (F) Model of a possible electron transfer path encoded by a representative multi-flavinylated gene cluster that includes a protein with a cell wall-binding domain. (G) Model of a possible electron transfer path encoded by a representative multi-flavinylated gene cluster that includes multiple Fr-like domains. (H) Model of possible bifurcated electron transfer pathway encoded by an RNF complex with a multi-flavinylated/multi-heme cytochrome RnfG.

electron transfer in multi-flavinylated proteins may thus represent an interesting subject for future studies.

## Materials and methods

### Genome collection

The 30,238 bacterial and 1672 archaeal genomes from the GTDB (release 05-RS95 of July 17, 2020) were downloaded with the taxonomy and the predicted protein sequences of the genomes (*Parks et al., 2020*).

### Functional annotation

Protein sequences were functionally annotated based on the accession of their best Hmmsearch match (version 3.3) (E-value cut-off 0.001) (*Eddy, 1998*) against the KOfam database (downloaded on February 18, 2020) (*Aramaki et al., 2020*). Domains were predicted using the same Hmmsearch procedure against the Pfam database (version 33.0) (*Mistry et al., 2021*). SIGNALP (version 5.0) was run to predict the putative cellular localization of the proteins using the parameters -org arch in archaeal genomes and -org gram +in bacterial genomes (*Almagro Armenteros et al., 2019*). Prediction of transmembrane helices in proteins was performed using TMHMM (version 2.0) (default parameters) (*Krogh et al., 2001*).

### Identification of flavinylated systems

The five genes downstream and upstream of an ApbE, FMN-binding domain, DUF3570 or DUF2271 encoding genes were first collected. Only gene clusters with at least one signal peptide or lipidation site predicted in one of the four target genes were considered for further analysis and were referred to as "flavinylation-associated gene clusters." The flavinylation-associated gene clusters were then assigned to 1 of the 10 flavinylated systems based on the presence of key genes reported in *Supplementary file 2*. The RNF system was considered present if a $Na^+$-translocating ferredoxin: $NAD^+$ oxidoreductase subunit B (RnfB, KEGG accession K03616) was encoded within a flavinylation-associated gene cluster. The nitrous oxide reduction system (Nos) was considered present if a nitrous oxide reductase (NosZ, KEGG accession K00376) was encoded within a flavinylation-associated gene cluster. The organohalide respiration system was defined by the presence of a PceA enzyme (Pfam accession PF13486) encoded within a flavinylation-associated gene cluster. The extracellular electron transfer system was considered present if a NADH dehydrogenase (KEGG accession K03885) with a transmembrane helix was encoded within a flavinylation-associated gene cluster. The NQR was defined by the $Na^+$-transporting NADH:ubiquinone oxidoreductase subunit F (NqrF, KEGG accession K00351) encoded within a flavinylation-associated gene cluster. The NapH-like was considered present if an 'NapH-like' iron-sulfur cluster-binding protein (Pfam accession PF12801 or KEGG accession K19339) was encoded within a flavinylation-associated gene cluster. NapH-like gene clusters that encoded a NirS (KEGG accession K15864) were identified as containing a nitrite reductase. NapH-like gene clusters that encoded an ExaA enzyme (KEGG accession K00114) were identified as containing an alcohol oxidase. The MsrQ-like system was defined by the presence of 'MsrQ-like' (Pfam accession PF01794) gene within a flavinylation-associated gene cluster. The PepSY-like system was defined by a PepSY-like (Pfam accessions PF03929 and PF16357) protein encoded within a flavinylation-associated gene cluster. The DsbD system was considered present if a DsbD protein (Pfam accession PF02683) was encoded within a flavinylation-associated gene cluster. Finally, the NQR/RNF-like system was considered present if a flavinylation-associated gene cluster encoded a protein with an N-terminal membrane domain that is homologous to NqrB/RnfD (Pfam

accession PF03116) and a cytosolic C-terminal domain homologous to the NAD-binding domain NqrF (Pfam accession PF00175).

## Phylogenetic analyses of the 'NapH-like' iron-sulfur cluster-binding protein sequences

The 'NapH-like' iron-sulfur cluster-binding protein tree was built as follows. Sequences were aligned using MAFFT (version 7.390) (–auto option) (*Katoh and Standley, 2016*). The alignment was further trimmed using Trimal (version 1.4.22) (–gappyout option) (*Capella-Gutierrez et al., 2009*). Tree reconstruction was performed using IQ-TREE (version 1.6.12) (*Nguyen et al., 2015*), using Model-Finder to select the best model of evolution (*Kalyaanamoorthy et al., 2017*) and with 1000 ultrafast bootstrap (*Hoang et al., 2018*).

## Concatenated 16 ribosomal proteins phylogeny

A maximum-likelihood tree was calculated based on the concatenation of 14 ribosomal proteins (L2, L3, L4, L5, L6, L14, L15, L18, L22, L24, S3, S8, S17, and S19). Homologous protein sequences were aligned using MAFFT (version 7.390) (–auto option) (*Katoh and Standley, 2016*) and alignments refined to remove gapped regions using Trimal (version 1.4.22) (–gappyout 570 option) (*Capella-Gutierrez et al., 2009*). The protein alignments were concatenated with a final alignment of 9152 genomes and 2850 positions. Tree reconstruction was performed using IQ-TREE (version 1.6.12) (*Nguyen et al., 2015*). A LG + I + G4 model of evolution was selected using ModelFinder (*Kalyaanamoorthy et al., 2017*) and 1000 ultrafast bootstraps were performed (*Hoang et al., 2018*).

## DUF2271 and DUF3570 sequence analyses

Sequences of flavinylation-associated DUF3570 and DUF2271 proteins were aligned using EMBL-EBI Clustal Omega Multiple Sequence Alignment (*Madeira et al., 2019*). Sequence logos of the flavinylation sites shown in *Figure 3* were generated in R using the 'ggseqlogo' package (*Wagih, 2017*).

## DUF2271 and DUF3570 overexpression and purification

A synthetic construct of the signal peptide-truncated *A. ursilacus* IGB-41 DUF2271 gene (NCBI accession WP_053936890.1) was subcloned into the pMCSG53 vector. A second construct contained a ribosome-binding site and the signal peptide-truncated cognate *apbE* (NCBI accession WP_053936888.1) just downstream of the DUF2271 gene. A similar cloning strategy was used for the *C. luteolum* DSM 273 DUF3570 (NCBI accession ABB24424.1) and its cognate *apbE* (NCBI accession ABB24423.1). Point mutations of the DUF3570- and DUF2271-encoding genes were generated using the NEB Q5 Site-Directed Mutagenesis Kit. Briefly, overlapping primers containing mutated sequences were used in a PCR using pMCSG53::DUF3570 or pMCSG53::DUF2771 as DNA template to generate expression vectors containing respective mutant sequences. Plasmids containing wild-type protein sequences were removed using digestion with the DpnI enzyme, which only acts on methylated DNA sequences.

Sequence verified plasmids were transformed in *E. coli* BL21 cells (Rosetta DE3, Novagen). A single colony of each expression strain was isolated on Luria-Bertani (LB) agar supplemented with carbenicillin (100 µg/mL) and inoculated into 15 mL of LB. Following overnight growth, cultures were diluted in 500 mL of brain heart infusion broth to a final OD600 of 0.1. When the OD600 reached 0.7–1 protein overexpression was induced by adding isopropyl β-D-1-thiogalactopyranoside to a final concentration of 1 mM. The culture was then incubated overnight at 25°C with aeration and collected by centrifugation (7000 × g for 15 min). After removing the supernatant, cells were washed in 30 mL of lysis buffer (5:1 v/weight of pellet; 300 mM NaCl, 1 mM dithiothreitol, 10 mM imidazole, 1 mM ethylenediaminetetraacetic acid [EDTA], and 50 mM Tris-HCl pH = 7.5). Pelleted cells were stored at −80°C overnight, resuspended in lysis buffer, lysed by sonication (8 × 30 s pulses), and cleared by centrifugation (40,000 × g for 30 min).

For the purification of *A. ursilacus* DUF2271, cell lysate was collected and loaded onto a 5 mL His-TrapTM column (GE Healthcare) using the ÄKTA Pure FPLC. Protein was eluted using an imidazole concentration gradient with a maximal concentration of 500 mM. Protein concentrations of eluted

fractions containing His$_6$-tagged DUF2271 were measured on a DeNovix DS-11 FX+Spectrophotommeter based on protein molar mass and extinction coefficient and standardized to 0.4 mg/mL.

Initial observations revealed that the majority of expressed *C. luteolum* DUF3570 was present in the lysed cell pellet. To purify *C. luteolum* DUF3570, the cell pellet was washed with wash buffer (10:1 v/weight of pellet; 100 mM Tris-HCl pH = 7.5, 300 mM NaCl, 2 mM 2-mercaptoethanol, 1 M guanidine-HCl, 1 mM EDTA, and 2% w/v Triton X-100) and centrifuged at 40,000 × *g* for 30 min. The supernatant was then discarded and this washing step was repeated until the supernatant was clear. The cell pellet was then resuspended in wash buffer (10:1 v/weight of pellet) without guanidine-HCl and Triton X-100 and centrifuged (40,000 × *g* for 15 min) to remove guanidine-HCl and Triton X-100. After supernatant was discarded, the pellet was resuspended in extraction buffer (4:1 v/weight of pellet; 100 mM Tris-HCl pH = 7.5, 300 mM NaCl, 10 mM imidazole, 6 M guanidine-HCl, 2 mM 2-mercaptoethanol, and 1 mM EDTA) and protein was denatured by overnight rotator mixation. Supernatant containing denatured DUF3570 was collected by centrifugation (20,000 × *g* for 30 min).

Subsequent *C. luteolum* DUF3570 purification steps were conducted on a Ni-NTA column. Specifically, 4 mL of Ni-NTA slurry (Nuvia IMAC Resin, 25 mL) were added into a glass chromatography column (Econo-Column, 1.0 × 10 cm$^2$). The column was prepared with 10 mL of column wash buffer and the samples containing denatured DUF3570 were then loaded. Bound protein was eluted using 5 mL of modified column wash buffer containing 50, 100, 200, or 500 mM of imidazole. To prevent guanidine-HCl from forming precipitates with SDS in following steps, eluted samples were mixed with 100% ethanol (9:1 v/v) and incubated at −20℃ for 10 min. After centrifugation at 21,100 × *g* for 5 min and removal of the supernatant, the pelleted protein was washed once with 90% ethanol. The sample was centrifuged again and the pellet was resuspended in diH$_2$O. Protein concentrations of eluted fractions containing His$_6$-tagged DUF3570 were measured as described above and standardized to 1.2 mg/mL.

## DUF2271 and DUF3570 flavinylation analyses

Purified and normalized DUF2271 and DUF3570 were loaded and separated on a 12% Bis-Tris gel. Prior to gel staining, flavinylated bands were visualized under UV due to the UV resonance of the flavin molecule. Visualizations of Coomassie blue stained protein were captured with an iBright 1500 gel imager.

## Acknowledgements

We thank Daniel Portnoy and Dominique Missiakas for providing critical feedback on the manuscript. RM and JFB acknowledge funding support from the Chan Zuckerberg Biohub and the Innovative Genomics Institute at University of California, Berkeley. RR-L acknowledges funding support from the National Academies of Sciences, Engineering, and Medicine (Ford Foundation Fellowship). SHL acknowledges funding support from the National Institute of Allergy and Infectious Diseases of the National Institutes of Health (K22 AI144031).

## Additional information

### Competing interests

Jillian F Banfield: is a founder of Metagenomi. The other authors declare that no competing interests exist.

### Funding

| Funder | Grant reference number | Author |
| --- | --- | --- |
| National Institute of Allergy and Infectious Diseases | K22 AI144031 | Samuel H Light |
| Ford Foundation | | Rafael Rivera-Lugo |
| Chan Zuckerberg Initiative | | Raphaël Méheust<br>Jillian F Banfield |

Innovative Genomics Institute

Raphaël Méheust
Jillian F Banfield

The funders had no role in study design, data collection and interpretation, or the decision to submit the work for publication.

## Author contributions
Raphaël Méheust, Conceptualization, Investigation, Writing - original draft; Shuo Huang, Formal analysis, Investigation, Visualization, Writing - review and editing; Rafael Rivera-Lugo, Investigation, Writing - review and editing; Jillian F Banfield, Supervision, Funding acquisition, Writing - review and editing; Samuel H Light, Conceptualization, Funding acquisition, Writing - original draft

## Author ORCIDs
Raphaël Méheust (iD) https://orcid.org/0000-0002-4847-426X
Rafael Rivera-Lugo (iD) http://orcid.org/0000-0002-2346-2297
Samuel H Light (iD) https://orcid.org/0000-0002-8074-1348

## Decision letter and Author response
Decision letter https://doi.org/10.7554/eLife.66878.sa1
Author response https://doi.org/10.7554/eLife.66878.sa2

# Additional files

## Supplementary files
• Supplementary file 1. Annotations of the flavinylation-associated gene clusters discussed in the study. Gene clusters of the 10 flavinylated systems discussed in this study (RNF, NQR, Nos, extracellular electron transfer, organohalide respiration, NapH-like, MsrQ-like, NQR/RNF-like, DsbD/DsbD-like, and PepSY-like), the P19 system, DUF2271, DUF3570, and multi-flavinylated proteins. The five genes downstream and upstream of each key gene (column K) were identified and annotated using signal peptide prediction (column G), transmembrane helix prediction (column H), PFAM (column I), and KEGG databases (column J). Genome accession and taxonomy are presented in columns A and B, respectively. Open reading frame (ORF) accessions are shown in column C. Columns D and E correspond to the scaffold accession and the ORF coordinates on the scaffold (start, end, and strand). Column F indicates the size of ORFs in amino acids.

• Supplementary file 2. Summary of prevalence of characterized and uncharacterized systems in prokaryotic genomes.

• Supplementary file 3. Fumarate reductase-like associations with flavin mononucleotide (FMN)-binding domains and cytochromes.

• Transparent reporting form

## Data availability
All data generated or analysed during this study are included in the manuscript and supporting files.

The following previously published dataset was used:

| Author(s) | Year | Dataset title | Dataset URL | Database and Identifier |
|---|---|---|---|---|
| Parks D, Chuvochina, Waite, Rinke, Skarshewski, Chaumeil, Hugenholtz | 2019 | A standardized bacterial taxonomy based on genome phylogeny substantially revises the tree of life (release R04-RS89) | https://data.gtdb.ecogenomic.org/releases/release89/ | NCBI/GTDB, 30148503 |

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
