## [Decision Letter]

**Acceptance summary:**

Light and coworkers provide evidence from mining 31,910 prokaryotic genomes for the widespread occurrence of extracytosolic flavinylated FMN-binding domains in bacteria. They discovered extracytosolic flavinylation of five protein classes potentially involved in transmembrane electron transfer. The study also proposes new connections between respiration and iron assimilation and identifies two novel substrates of ApbE enzymes. This work should inspire further work in the fields of redox enzymology and bioenergetics to characterize the suggested involvement of flavinylated protein complexes in prokaryotes.

**Decision letter after peer review:**

Thank you for submitting your article "Widespread bacterial protein flavinylation in functionally distinct extracytosolic redox biochemistries" for consideration by *eLife*. Your article has been reviewed by 4 peer reviewers, including Pimchai Chaiven as the Reviewing Editor and Reviewer #1, and the evaluation has been overseen by Philip Cole as the Senior Editor. The following individuals involved in review of your submission have agreed to reveal their identity: Willem van Berkel (Reviewer #2); Peter Macheroux (Reviewer #3).

Essential revisions:

1. The introduction lacks a clear explanation about the mode of flavinylation of the FMN-binding proteins and how this relates to other covalent flavinylation systems (where an increase in redox potential of the flavin is a prominent effect of covalent binding). It is also not clearly explained whether the predicted flavinylation of the phosphate moiety of FMN is reversible.

2. Results and Discussion: The electron transfer properties of flavoproteins are not well explained. Quite some flavoproteins (e.g. flavodoxins) mediate one-electron transfer processes, and this is most likely the preferred way in the discussed transmembrane electron transport systems.

3. As mentioned by the authors, about 50% of the prokaryotic genomes analyzed harbor targets for flavinylation/and the FMN transferase. However, no discussion and not even a hint is provided what these 50% of prokaryotes have in common and what distinguishes this group from the other (50%) prokaryotes. Is it lifestyle (environment), energy production,.…?

4. On the other hand, the presented study leaves many issues unmentioned creating the (false) impression that all it takes to transport electrons across the membrane is a series of hemes and/or flavins along the way. For example, in the discussion of the very interesting hypothesis that flavinylation might replace multi-heme cytochromes under iron deficiency, discussed on page 20 (last para), the authors mention that "flavins possess two-electron transferring properties (ref. 46)" in contrast to the heme system. If this were true, the switch from heme to flavin would also imply that the electron transport itself would have to change from one- electron to two-electron transport. It is unclear that this would be compatible with all other components of the electron transport system. On the other hand, flavins can also – under certain circumstances and in certain environments – carry out one-electron transfer processes, e. g. DNA-photolyases, flavodoxins, etc. Thus, it is conceivable that the flavins operating in the suggested systems in prokaryotes also perform one-electron transport, similar to the operating mode of heme cytochromes. It is clear that we currently lack the biochemical/physical information to know what is really going on, but at least it should be discussed more thoroughly.

5. Title and wherever applicable: replace "biochemistries", "redox biochemistries", "uncharacterized biochemistries" and "cytosolic biochemistry" by "redox reactions/systems/processes" or "biochemical reactions" (Abstract, Introduction, pages 17 and 19). In our understanding "biochemistry" is a science and does not "exist" in plural.

6. Abbreviations: Explain protein (gene) abbreviations. Several abbreviations are used without proper introduction or explanation. I think it would enhance the readability, if these were properly explained, e. g. NQR, RNF, NapH-like, PceC, NosZ, PceA, etc.

7. Some adjustment and rearrangement of the introduction is required to improve the readability.

8. On page 2, line 56-58, the authors state that "…more recently…" but no reference is given here.

9. Issues regarding functions and properties of flavinylation.

9.1 What are the physiological functions of flavinylation of these proteins? Without having FMN flavinylated, many flavoenzymes can also mediate electron transfer. If these proteins can also bind FMN as a prosthetic group without flavinylation, one can address this issue by comparing redox properties of the enzyme-bound FMN with or without covalent linkage. Properties which should be characterized include redox potential measurement and reduction titration to investigate ability of these proteins to stabilize flavin semiquinones. One can address how does the binding of the flavin affects the redox potential (this is very important in order to understand the direction of electron transport).

9.2 In contrast to other covalent flavin attachments, the flavinylation addressed in the current work is reversible. Is anything known about the removal of flavins from the protein complexes in question?

How sure is it that the flavin is always covalently bound and what would be the consequence if this is not the case? Might there be next to iron limitation, also flavin limitation? Is the reversibility of flavinylation used for the overall regulation of electron transport?

9.3 Are there any enzymes that carry out de-flavinylation? If so, how are they regulated?

9.4 Is there any protein structural information about this mode of flavinylation available? For instance, is the flavin hidden in the protein or accessible? Do the amino acid sequence results can explain in more general terms the site(s) of flavinylation?

What is known about the environment of the flavin(s)? This can address if it "behaves" like a "free" flavin?

9.5 How sure is it that the conserved motif always represents covalent flavinylation?

9.6 Page 4, considerable amount of these proteins (50%) may remain inside the cytosol. Any ideas about their functional roles? Can we say more about the comparison with thioredoxins and cytochromes when we look at the 50% of bacteria that do not contain the flavinylation domains?

10. Page 6, can these DUF2271 or DUF3570 also be flavinylated by FAD or riboflavin?

11. Page 8, line 126, the sentence "…we found that a threonine to alanine mutation at the predicted…". This should be rephrased: "…we found that a threonine to alanine replacement/exchange at the predicted….".

12. Page 9, top sentences, the authors mention that 6,366 genomes contain transmembrane electron transfer apparatuses. It would be more informative to also mention the number of genomes without putative membrane-bound clusters.

13. Page 13: "NqrB/RnfD is flavinylated by ApbE and plays a role in electron transfer across the membrane" – is that a fact (reference?) or a hypothesis?

14. Page 15: correct mu(l)ti-heme in line 320.

15. Page 20, Discussion, in addition to the issue of Fe availability, these flavinylated proteins may provide other functional advantage for cells to use them in mediating electron transfer. Knowing their redox properties (Issue#1) would give a clue on this.

16. DUF2271 or DUF3570 obviously should be able to bind to FMN. Is it possible to model the three dimensional structures of these proteins and analyze the difference and similarity compared to the known FMN-binding domain? Do they use different scaffold to interact with FMN?

17. Discussion, what strategies do the authors have in mind for testing the proposed functions of the flavinylated domains? I would like to see more discussion about how to validate the proposed functions of the flavinylated domains.

I assume that most of the questions cannot be satisfactorily answered yet, but I think these issues should at least be addressed in the discussion in order to stress the need for further in depths biochemical studies that target the obvious complexity of these systems.

18. Figures 1, 4, 5. 6 – it is unclear how the structure of the systems presented in Figures 1, 4, 5 and 6 have been determined. Are these hypothetical? Taken from the literature? If these are taken from the literature, references should be provided in the figure captions.

19. Figure 1 – some of the text may be too small and hard to read; the text inside the genes with a dark color is not readable.

20. Figure 4 – some of the text is too small, hard to read.

21. Figure 4 presents four putative apparatuses. The authors should carry out experimental validation to confirm what these apparatuses are doing.

*Reviewer #1 (Recommendations for the authors):*

Issues that need clarification:

1. What are the physiological functions of flavinylation of these proteins? Without having FMN flavinylated, many flavoenzymes can also mediate electron transfer. If these proteins can also bind FMN as a prosthetic group without flavinylation, one can address this issue by comparing redox properties of the enzyme-bound FMN with or without covalent linkage. Properties which should be characterized include redox potential measurement and reduction titration to investigate ability of these proteins to stabilize flavin semiquinones.

2. Page 4, considerable amount of these proteins (50%) may remain inside the cytosol. Any ideas about their functional roles?

3. Page 6, can these DUF2271 or DUF3570 also be flavinylated by FAD or riboflavin?

4. Page 9, top sentences, the authors mention that 6,366 genomes contain transmembrane electron transfer apparatuses. It would be more informative to also mention the number of genomes without putative membrane-bound clusters.

5. Page 20, Discussion, in addition to the issue of Fe availability, these flavinylated proteins may provide other functional advantage for cells to use them in mediating electron transfer. Knowing their redox properties (Issue#1) would give a clue on this.

6. DUF2271 or DUF3570 obviously should be able to bind to FMN. Is it possible to model the three dimensional structures of these proteins and analyze the difference and similarity compared to the known FMN-binding domain? Do they use different scaffold to interact with FMN?

*Reviewer #2 (Recommendations for the authors):*

Title and wherever applicable: replace biochemistries with processes.

Abbreviations: Explain protein (gene) abbreviations.

Some adjustment and rearrangement of the introduction is required to improve the readability.

What strategies do the authors have in mind for testing the proposed functions of the flavinylated domains? I would like to see more discussion about how to validate the proposed functions of the flavinylated domains.

*Reviewer #3 (Recommendations for the authors):*

Personally, I would prefer replacing "redox biochemistries", "uncharacterized biochemistries" and "cytosolic biochemistry" by "redox reactions/systems/processes" or "biochemical reactions" (Abstract, Introduction, pages 17 and 19). In my understanding "biochemistry" is a science and does not "exist" in plural.

Several abbreviations are used without proper introduction or explanation. I think it would enhance the readability, if these were properly explained, e. g. NQR, RNF, NapH-like, PceC, NosZ, PceA, etc. especially in the absence of a list of abbreviations, which is obviously not required by the journal (?).

On page 2, line 56-58, the author state that "…more recently…" but no reference is given here.

On page 8, line 126, the sentence "…we found that a threonine to alanine mutation at the predicted…". This should be rephrased: "…we found that a threonine to alanine replacement/exchange at the predicted….".

On page 13: "NqrB/RnfD is flavinylated by ApbE and plays a role in electron transfer across the membrane" – is that a fact (reference?) or a hypothesis?

On page 15: correct mu(l)ti-heme in line 320.

*Reviewer #4 (Recommendations for the authors):*

The authors conducted a very systematic study of a large number of genomes. The study is carefully conducted, but throughout the manuscript, only speculative descriptions are provided since no hard claims are possible due to the lack of experimental data.

The main limitation of the study is that there is very limited validation of theoretical predictions. It is therefore very difficult to judge if the conclusions presented in this manuscript are correct or not. The authors themselves admit that observations are preliminary (p. 19). A high level of uncertainty can be found also in the Discussion part of this article.

There is only one experiment reported in this study, verification that the protein DUF2271 is flavinated in the presence of its cognate ApbE.

Specific comments:

Figures 1, 4, 5. 6 – it is unclear to me how the structure of the systems presented in Figures 1, 4, 5 and 6 have been determined. Are these hypothetical? Taken from the literature? If these are taken from the literature, references should be provided in the figure captions.

Figure 1 – some of the text may be too small and hard to read; the text inside the genes with a dark colour is not readable.

Figure 4 – some of the text is maybe too small, hard to read

Figure 4 presents four putative apparatuses. The authors should carry out experimental validation to confirm what these apparatuses are doing.

---

## [Author Response]

Essential revisions:1. The introduction lacks a clear explanation about the mode of flavinylation of the FMN-binding proteins and how this relates to other covalent flavinylation systems (where an increase in redox potential of the flavin is a prominent effect of covalent binding). It is also not clearly explained whether the predicted flavinylation of the phosphate moiety of FMN is reversible.

We’ve extensively revised the introduction to provide more background and address these points.

2. Results and Discussion: The electron transfer properties of flavoproteins are not well explained. Quite some flavoproteins (e.g. flavodoxins) mediate one-electron transfer processes, and this is most likely the preferred way in the discussed transmembrane electron transport systems.

We’ve extensively revised the introduction and Discussion sections to better address these points.

3. As mentioned by the authors, about 50% of the prokaryotic genomes analyzed harbor targets for flavinylation/and the FMN transferase. However, no discussion and not even a hint is provided what these 50% of prokaryotes have in common and what distinguishes this group from the other (50%) prokaryotes. Is it lifestyle (environment), energy production,.…?

It’s an interesting point and could greatly enhance our understanding of the function and significance of flavinylation. Unfortunately, we have not been able to identify features that clearly distinguish microbes that possess flavinylation-associated genes from those that do not. In reference to the related reviewer point, we performed additional analyses, but found no correlation between the presence of flavinylation-associated genes and cytochromes or thioredoxins within a genome. We’ve also been unsuccessful in our attempts to identify distinct habitats that distinguish microbes that encode flavinylation components. The difficulty in identifying features that differentiate these microbes is likely due to the multiple functional roles of flavinylation. While flavinylation is implicated in diverse bioenergetic activities, it has also been linked to other functions. For example, RNF is the most widespread flavinylated system and is used by some microbes for flavin bifurcation-based bioenergetic metabolism but by others for ferredoxin reduction (through reverse electron flow). Thus, while RNF is the main apparatus for flavin bifurcation-based bioenergetic metabolisms, its presence in a genome is not particularly predictive of whether or not a microbe uses this metabolic strategy. Another example of the functional heterogeneity of flavinylation is provided by Actinobacteria. While flavinylated systems linked to bioenergetic processes are distributed through most of the phylogenetic tree, almost all Actinobacteria possess a single MsrQ-like flavinylation system that likely functions in iron assimilation, implying a distinct significance of flavinylation in this context (Figure 2). The diverse functions of flavinylation thus likely mask features shared by microbes that possess it.

4. On the other hand, the presented study leaves many issues unmentioned creating the (false) impression that all it takes to transport electrons across the membrane is a series of hemes and/or flavins along the way. For example, in the discussion of the very interesting hypothesis that flavinylation might replace multi-heme cytochromes under iron deficiency, discussed on page 20 (last para), the authors mention that "flavins possess two-electron transferring properties (ref. 46)" in contrast to the heme system. If this were true, the switch from heme to flavin would also imply that the electron transport itself would have to change from one- electron to two-electron transport. It is unclear that this would be compatible with all other components of the electron transport system. On the other hand, flavins can also – under certain circumstances and in certain environments – carry out one-electron transfer processes, e. g. DNA-photolyases, flavodoxins, etc. Thus, it is conceivable that the flavins operating in the suggested systems in prokaryotes also perform one-electron transport, similar to the operating mode of heme cytochromes. It is clear that we currently lack the biochemical/physical information to know what is really going on, but at least it should be discussed more thoroughly.

Good point about the one- versus two-electron transferring properties. We’ve removed the mention of two-electron transferring properties in relation to multi-flavinylated proteins, modified the introduction to provide additional context about one-electron transport, and made extensive revisions to, hopefully, better contextualize the findings.

5. Title and wherever applicable: replace "biochemistries", "redox biochemistries", "uncharacterized biochemistries" and "cytosolic biochemistry" by "redox reactions/systems/processes" or "biochemical reactions" (Abstract, Introduction, pages 17 and 19). In our understanding "biochemistry" is a science and does not "exist" in plural.

Good point. We’ve changed the title and removed this usage throughout the manuscript.

6. Abbreviations: Explain protein (gene) abbreviations. Several abbreviations are used without proper introduction or explanation. I think it would enhance the readability, if these were properly explained, e. g. NQR, RNF, NapH-like, PceC, NosZ, PceA, etc.

Thanks for pointing that out. We’ve provided more background for everything referenced. As another approach to reducing the confusion around referenced genes, we’ve moved some of the gene references to a section in the methods that defines the criteria used to identify each system.

7. Some adjustment and rearrangement of the introduction is required to improve the readability.

We’ve extensively revised (and, hopefully, improved) the introduction.

8. On page 2, line 56-58, the authors state that "…more recently…" but no reference is given here.

Citations added. Thanks for pointing out the omission.

9. Issues regarding functions and properties of flavinylation.9.1 What are the physiological functions of flavinylation of these proteins? Without having FMN flavinylated, many flavoenzymes can also mediate electron transfer. If these proteins can also bind FMN as a prosthetic group without flavinylation, one can address this issue by comparing redox properties of the enzyme-bound FMN with or without covalent linkage. Properties which should be characterized include redox potential measurement and reduction titration to investigate ability of these proteins to stabilize flavin semiquinones. One can address how does the binding of the flavin affects the redox potential (this is very important in order to understand the direction of electron transport).

Proteins flavinylated by ApbE have low affinity for flavins (Borshchevskiy et al., 2015). Consistent with this previous finding, we and others have found that flavinylated proteins do not copurify with non-covalently bound flavin when expressed in the absence of ApbE (Barquera et al., 2001). By contrast, ApbE does copurify with FAD (it’s substrate) (Boyd et al., 2011). Presumably, ApbE binds the flavin substrate and the [S/T]GA[S/T] motif from the substrate protein. As such, non-covalent flavin binding to the substrate protein would likely competitively inhibit flavinylation, which may provide a rationale for the low affinity.

At any rate, it’s generally accepted that the covalent linkage to FMN-binding proteins (or at least the best-studied protein, NqrC) does not impact the redox potential of the flavin. Also, others have found evidence that the semiquinone is stabilized in flavinylated proteins (Backiel et al., 2008; Barquera et al., 2006) – presumably by non-covalent interactions with the protein. We have added information about these properties to the introduction section.

9.2 In contrast to other covalent flavin attachments, the flavinylation addressed in the current work is reversible. Is anything known about the removal of flavins from the protein complexes in question?

To the best of our knowledge, ApbE catalyzed flavinylation is irreversible. We are not aware of a “de-flavinylating” enzyme.

How sure is it that the flavin is always covalently bound and what would be the consequence if this is not the case? Might there be next to iron limitation, also flavin limitation? Is the reversibility of flavinylation used for the overall regulation of electron transport?

We hesitate to make too sweeping a statement, but think the flavin is probably typically covalently bound and that control of the state of the post-translational modification is unlikely to serve a regulatory function. Consistent with this idea, our previous proteomics experiments on *Listeria monocytogenes* found that >99% of the motif-containing peptides were flavinylated (Light et al., 2018).

The point about flavin limitation is interesting. Obviously, flavin differs from iron in that it can be synthesized de novo and so isn’t a scarce environmental resource, at least not in the same sense. On the other hand, environmental flavins may be readily available in certain environments rich in organic matter and microbes that reside in such environments are often flavin auxotrophs. An environmental abundance of flavins may lead to a decreased demand for energy intensive flavin biosynthesis that, similar to the lack of environmental iron, changes the flavin vs iron usage cost/benefit ratio.

9.3 Are there any enzymes that carry out de-flavinylation? If so, how are they regulated?

We are not aware of such an enzyme.

9.4 Is there any protein structural information about this mode of flavinylation available? For instance, is the flavin hidden in the protein or accessible? Do the amino acid sequence results can explain in more general terms the site(s) of flavinylation?What is known about the environment of the flavin(s)? This can address if it "behaves" like a "free" flavin?

To the best of our knowledge, the only structures that exist of AbpE-flavinylated proteins are subunits of the NQR complex (Borshchevskiy et al., 2015; Steuber et al., 2014). The NQR complex from the bacterium *Vibrio harveyi* contains two flavinylation sites separately located on the NqrB and NqrC subunits. NqrC contains an FMN-binding domain, while NqrB does not (it has an unrelated domain found in NQR and RNF complexes). The NqrC structure is thus likely more relevant for understanding the majority of flavinylated proteins discussed in our study. Within the NqrC structure, the [S/T]GA amino acids within the [S/T]GA[S/T] motif are on a loop and the flavinylated S/T (the second S/T in the motif) is at the start of an alpha-helix. The isoalloxazine ring within the covalently bound flavin forms hydrogen bonds with the NqrC main chain and a threonine side chain. The more hydrophobic portion of the ring is positioned between two neighboring subunits (NqrD and NqrF) within the multi-subunit complex. Given the importance of the interactions with these NQR-specific subunits, it’s difficult to predict what the flavinylation site would look like in other flavinylated proteins.

As explained above, several previous studies have presented evidence that the semiquinone is stabilized in the bound NqrC (even the absence of other NQR subunits) and that this likely results from non-covalent protein interactions rather than the covalent linkage. We now discuss this point in the introduction section.

9.5 How sure is it that the conserved motif always represents covalent flavinylation?

The [S/T]GA[S/T] sequence is not a particularly stringent “motif”and occurs in numerous proteins. This sequence is unlikely to be sufficient for flavinylation. However, several lines of evidence suggest that the conserved motif *within the identified domains* is typically (if not always) flavinylated: (1) The sequence conservation of the motif (which, as noted above, does not seem to be involved in binding the flavin non-covalently). (2) The presence of the motif within a number of protein domains that have been established to be flavinylated. (3) The co-localization of the genes encoding ApbEs and FMN-binding domains, DUF2271s, and DUF3570s on gene clusters.

On the other hand our analysis likely overlooks biologically relevant flavinylation sites that occur outside of the identified domains. For example, the fumarate reductase that we recently showed was flavinylated in *Listeria monocytogenes* contains a conserved flavinylation motif that is not embedded within an identified domain (Light et al., 2019). In general, presently unknown structural features outside the [S/T]GA[S/T] sequence are likely essential for flavinylation.

9.6 Page 4, considerable amount of these proteins (50%) may remain inside the cytosol. Any ideas about their functional roles? Can we say more about the comparison with thioredoxins and cytochromes when we look at the 50% of bacteria that do not contain the flavinylation domains?

Of the 19,203 genomes that we identified as encoding ApbE and/or FMN-binding domains 15,158 were computationally predicted to contain a protein with a signal peptide and thus likely possess extracytosolic flavinylation. This probably underestimates the number of genomes that encode extracytosolic proteins, because the signal peptide prediction tool appears to be fairly conservative (i.e., misses some that must be extracytosolic). While cytosolic flavinylation undoubtedly occurs, it is likely less common than extracytosolic flavinylation.

We considered including analyses of flavinylated cytosolic proteins, but ultimately decided that it would detract from the main focus of the paper. Many of the cytosolic proteins are likely involved in a mechanistically similar cytosolic electron transfer. FMN-binding domains are frequently associated with cytosolic enzymes that contain fumarate reductase-like domains. Other proteins with FMN-binding domains are annotated as flavoproteins or have predicted oxidoreductase activities.

It is an interesting point about the thioredoxins and cytochromes. Given the evidence of functional redundancy of flavinylation with these other redox-active proteins, it would make sense if flavinylation “replaced” cytochromes or thioredoxins in some genomes. To address this point, we have performed additional analyses to address whether genomes with flavinylation components are more or less likely to possess genes that encode thioredoxins or cytochromes. We compared genomes that contain or lack flavinylation-associated components, but did not find a significant difference in cytochrome or thioredoxin gene content. There may be too many other variables to uncover a relationship between flavinylation and these other redox proteins in this way.

10. Page 6, can these DUF2271 or DUF3570 also be flavinylated by FAD or riboflavin?

Not to our knowledge, though we only observed the recombinantly expressed proteins in the presence or absence of ApbE. We performed MALDI TOF analysis of the DUF2271 (but were unable to analyze the DUF3570 protein due to technical issues). The detected ions are consistent with the protein being unflavinylated in the absence of ApbE and a single FMN being covalently attached in the presence of ApbE. In addition, a number of ApbEs from different organisms have been previously biochemically characterized and, without exception, been found to catalyze the same FMNylation reaction. For these reasons, we are confident FMNylation is the biologically relevant post-translational modification.

11. Page 8, line 126, the sentence "…we found that a threonine to alanine mutation at the predicted…". This should be rephrased: "…we found that a threonine to alanine replacement/exchange at the predicted….".

Good point. Fixed.

12. Page 9, top sentences, the authors mention that 6,366 genomes contain transmembrane electron transfer apparatuses. It would be more informative to also mention the number of genomes without putative membrane-bound clusters.

Good point. This number (2,595) has been added to the text. Some of these microbes may use described electron transfer apparatuses encoded elsewhere on the genome (i.e., not in the same gene cluster as apbE or the other domains). As referenced in the text, this is likely true of some of the microbes with the Dsb/DUF3570 system. It’s also likely that we have failed to identify less common electron transfer apparatuses that account for electron transfer in some microbes.

13. Page 13: "NqrB/RnfD is flavinylated by ApbE and plays a role in electron transfer across the membrane" – is that a fact (reference?) or a hypothesis?

There’s pretty strong (but not definitive) evidence to support that conclusion. We’ve changed the language to reflect this uncertainty and added a reference that supports the statement.

14. Page 15: correct mu(l)ti-heme in line 320.

Thanks. Fixed.

15. Page 20, Discussion, in addition to the issue of Fe availability, these flavinylated proteins may provide other functional advantage for cells to use them in mediating electron transfer. Knowing their redox properties (Issue#1) would give a clue on this.

We agree that iron limitation is unlikely to be the only advantage/reason for the existence of flavinylation. We’ve also expanded the discussion of what is known about the redox potential of ApbE-flavinylated proteins (i.e., that the semiquinone state is stabilized). We’re not experts in molecular-level electron transfer and may still not fully appreciate the functional implications of the characterized redox properties. Any additional reviewer input on these points would be appreciated.

16. DUF2271 or DUF3570 obviously should be able to bind to FMN. Is it possible to model the three dimensional structures of these proteins and analyze the difference and similarity compared to the known FMN-binding domain? Do they use different scaffold to interact with FMN?

Unfortunately, structures of proteins with sequence homology have not been reported. We’re attempting to crystallize these proteins and, if successful, should gain insight into the structural similarities/differences between flavinylated proteins.

17. Discussion, what strategies do the authors have in mind for testing the proposed functions of the flavinylated domains? I would like to see more discussion about how to validate the proposed functions of the flavinylated domains.I assume that most of the questions cannot be satisfactorily answered yet, but I think these issues should at least be addressed in the discussion in order to stress the need for further in depths biochemical studies that target the obvious complexity of these systems.

We’ve made extensive changes to the Discussion section and now emphasize the need for additional studies to address the mechanism and function of the systems described in the manuscript. We decided against providing too much detail about our future experimental plans (which may or may not produce useful results), but we’re currently focusing on several flavinylated systems that seem particularly interesting and potentially tractable. We’re developing genetic approaches that can be used to identify the function of these systems and recombinant protein expression that will support biochemical and structural studies.

18. Figures 1, 4, 5. 6 – it is unclear how the structure of the systems presented in Figures 1, 4, 5 and 6 have been determined. Are these hypothetical? Taken from the literature? If these are taken from the literature, references should be provided in the figure captions.

The models in figure 1 and most of figure 5 are based on previous findings. We have updated the figure legend to include appropriate references. Models in figures 4-6 that show hypothesized complexes and electron transfer pathways. Electron transfer hypotheses are based on (1) the content of identified gene clusters and (2) the established function of homologous proteins. We have rephrased these legends to clarify what the models represent.

19. Figure 1 – some of the text may be too small and hard to read; the text inside the genes with a dark color is not readable.

Thanks for pointing that out. We’ve increased the text size.

20. Figure 4 – some of the text is too small, hard to read.

Thanks for pointing that out. We’ve increased the text size.

21. Figure 4 presents four putative apparatuses. The authors should carry out experimental validation to confirm what these apparatuses are doing.

Most of the identified systems are in non-model and even uncultivated organisms. Many of the putative components (e.g., electron donor/acceptor) are unknown. Reconstituting and performing rigourous biochemical experiments on multi-subunit transmembrane protein complexes is challenging even in ideal circumstances. Experimentally validating the proposed systems would be a difficult and lengthy undertaking. As this type of validation couldn’t be completed in a timely manner, we think it would be better left for future studies.

References:

Backiel J, Juárez O, Zagorevski DV, Wang Z, Nilges MJ, Barquera B. 2008. Covalent binding of flavins to RnfG and RnfD in the Rnf complex from Vibrio cholerae. Biochemistry 47:11273–11284. doi:10.1021/bi800920j

Barquera B, Häse CC, Gennis RB. 2001. Expression and mutagenesis of the NqrC subunit of the NQR respiratory Na(+) pump from Vibrio cholerae with covalently attached FMN. FEBS Lett 492:45–49. doi:10.1016/s0014-5793(01)02224-4

Barquera B, Ramirez-Silva L, Morgan JE, Nilges MJ. 2006. A New Flavin Radical Signal in the Na -pumping NADH:Quinone Oxidoreductase from Vibrio cholerae. Journal of Biological Chemistry. doi:10.1074/jbc.m605765200

Borshchevskiy V, Round E, Bertsova Y, Polovinkin V, Gushchin I, Ishchenko A, Kovalev K, Mishin A, Kachalova G, Popov A, Bogachev A, Gordeliy V. 2015. Structural and functional investigation of flavin binding center of the NqrC subunit of sodium-translocating NADH:quinone oxidoreductase from Vibrio harveyi. PLoS One 10:e0118548. doi:10.1371/journal.pone.0118548

Boyd JM, Endrizzi JA, Hamilton TL, Christopherson MR, Mulder DW, Downs DM, Peters JW. 2011. FAD binding by ApbE protein from *Salmonella enterica*: a new class of FAD-binding proteins. J Bacteriol 193:887–895. doi:10.1128/JB.00730-10

Light SH, Méheust R, Ferrell JL, Cho J, Deng D, Agostoni M, Iavarone AT, Banfield JF, D’Orazio SEF, Portnoy DA. 2019. Extracellular electron transfer powers flavinylated extracellular reductases in Gram-positive bacteria. Proc Natl Acad Sci U S A 116:26892–26899. doi:10.1073/pnas.1915678116

Light SH, Su L, Rivera-Lugo R, Cornejo JA, Louie A, Iavarone AT, Ajo-Franklin CM, Portnoy DA. 2018. A flavin-based extracellular electron transfer mechanism in diverse Gram-positive bacteria. Nature 562:140–157. doi:10.1038/s41586-018-0498-z

Steuber J, Vohl G, Casutt MS, Vorburger T, Diederichs K, Fritz G. 2014. Structure of the V. cholerae Na+-pumping NADH:quinone oxidoreductase. Nature 516:62–67. doi:10.1038/nature14003